# Regularized Robustly Reliable Learners and Instance Targeted Attacks

## Abstract

Instance-targeted data poisoning attacks, where an adversary corrupts a training set to induce errors on specific test points, have raised significant concerns. Balcan et al. [2022] proposed an approach to addressing this challenge by defining a notion of *robustly-reliable learners* that provide per-instance guarantees of correctness under well-defined assumptions, even in the presence of data poisoning attacks. They then give a generic optimal (but computationally inefficient) robustly-reliable learner as well as a computationally efficient algorithm for the case of linear separators over log-concave distributions.

In this work, we address two challenges left open by Balcan et al. [2022]. The first is that the definition of robustly-reliable learners in Balcan et al. [2022] becomes vacuous for highly-flexible hypothesis classes: if there are two classifiers $h_0, h_1 \in \mathcal{H}$ both with zero error on the training set such that $h_0(x) \neq h_1(x)$, then a robustly-reliable learner must abstain on $x$. We address this problem by defining a modified notion of *regularized* robustly-reliable learners that allows for nontrivial statements in this case. The second is that the generic algorithm of Balcan et al. [2022] requires re-running an ERM oracle (essentially, retraining the classifier) on each test point $x$, which is generally impractical even if ERM can be implemented efficiently. To tackle this problem, we show that at least in certain interesting cases we can design algorithms that can produce their outputs in time sublinear in training time, by using techniques from dynamic algorithm design.

## 1 Introduction

As Machine Learning and AI are increasingly used for critical decision-making, it is becoming more important than ever that these systems be trustworthy and reliable. This means they should know (and say) when they are unsure, they should be able to provide real explanations for their answers and why those answers should be trusted (not just how the prediction was made), and they should be robust to malicious or unusual training data and to adversarial or unusual examples at test time.

Balcan et al. [2022] proposed an approach to addressing this problem by defining a notion of *robustly-reliable learners* that provide per-instance guarantees of correctness under well-defined assumptions, even in the presence of data poisoning attacks. This notion builds on the definition of *reliable learners* by Rivest and Sloan [1988]. In brief, a robustly-reliable learner $\mathcal{L}$ for some hypothesis class $\mathcal{H}$, when given a (possibly corrupted) training set $S'$, produces a classifier $\mathcal{L}_{S'}$ that on any example $x$ outputs both a prediction $y$ and a confidence level $k$. The interpretation of the pair $(y, k)$ is that $y$ is guaranteed to equal the correct label $f^*(x)$ if (a) the target function $f^*$ indeed belongs to $\mathcal{H}$ and (b) the set $S'$ contains at most $k$ corrupted points; here, $k < 0$ corresponds to abstaining. Balcan et al. [2022] then provide a generic pointwise-optimal algorithm for this problem: one that for each $x$ outputs the largest possible confidence level of any robustly-reliable learner. They also give efficient

algorithms for the case of homogeneous linear separators over uniform and log-concave distributions, as well as analysis of the probability mass of points for which it outputs large values of $k$.

In this work, we address two challenges left open by Balcan et al. [2022]. The first is that the definition of robustly-reliable learners in Balcan et al. [2022] becomes vacuous for highly-flexible hypothesis classes: if there are two classifiers $h_0, h_1 \in \mathcal{H}$ both with zero error on the training set such that $h_0(x) \neq h_1(x)$, then a robustly-reliable learner must abstain on $x$. We address this problem by defining a modified notion of *regularized* robustly-reliable learners that allows for nontrivial statements in this case. The second is that the generic algorithm of Balcan et al. [2022] requires re-running an ERM oracle (essentially, retraining the classifier) on each test point $x$, which is generally impractical even if ERM can be implemented efficiently. To tackle this problem, we show that at least in certain interesting cases we can design algorithms that can make predictions in time sublinear in training time, by using techniques from dynamic algorithm design, such as Bosek et al. [2014].

## 1.1 Main contibutions

Our main contributions are three-fold.

1. The first is a definition of a *regularized* robustly-reliable learner, and of the *region* of points it can certify, that is appropriate for highly-flexible hypothesis classes. We then analyze the largest possible set of points that any regularized robustly-reliable learner could possibly certify, and provide a *generic pointwise-optimal algorithm* whose regularized robustly-reliable region ($\text{R}^4$) matches this optimal set ($\text{OPTR}^4$).

2. The second is an analysis of the probability mass of this $\text{OPTR}^4$ set in some interesting special cases, proving sample complexity bounds on the number of training examples needed (relative to the data poisoning budget of the adversary and the complexity of the target function) in order for $\text{OPTR}^4$ to w.h.p. have a large probability mass.

3. Finally, the third is an analysis of efficient regularized robustly-reliable learning algorithms for interesting cases, with a special focus on algorithms that are able to output their reliability guarantees more efficiently than re-training the entire classifier. In one case we do this through a bi-directional dynamic programming algorithm, and in another case by utilizing algorithms for maximum matching that are able to quickly re-establish the maximum matching when a few nodes are added to or deleted from the graph.

In a bit more detail, for a given complexity (or "unnaturalness") measure $\mathcal{C}$, a regularized robustly-reliable learner $\mathcal{L}$ is given as input a possibly-corrupted training set $S'$ and outputs a function (an "extended classifier") $\mathcal{L}_{S'}$. The extended classifier $\mathcal{L}_{S'}$ takes in two inputs: a test example $x$ and a poisoning budget $b$, and outputs a prediction $y$ along with two complexity levels $c_{\text{low}}$ and $c_{\text{high}}$. The meaning of the triple $(y, c_{\text{low}}, c_{\text{high}})$ is that $y$ is guaranteed to be the correct label $f^*(x)$ if the training set $S'$ contains at most $b$ poisoned points and the complexity of the target function $f^*$ is less than $c_{\text{high}}$. Moreover, there should *exist* a classifier $f$ of complexity at most $c_{low}$ that makes at most $b$ mistakes on $S'$ and has $f(x) = y$. Thus, if we, as a user, believe that a complexity at or above $c_{high}$ is "unnatural" and that the training set should contain at most $b$ corrupted points, then we can be confident in the predicted label $y$. We then analyze the set of points for which $c_{low} \leq c < c_{high}$ for a given complexity level $c$, and show there exists an algorithm that is simultaneously optimal in terms of the size of this set for all values of $c$.

The above description has been treating the complexity function $\mathcal{C}$ as a data-independent quantity. However, in many cases we may want to consider notions of "unnaturalness" that involve how the classifier relates to the test point, the training examples, or both. For instance, if $x$ is surrounded by positive examples, we might view a positive classification as more natural than a negative one even if we allow arbitrary functions as classifiers; one way to model this would be to define the complexity of a classifier $h$ with respect to test point $x$ as $1/r(h, x)$ where $r(h, x)$ is the distance of $x$ to $h$'s decision boundary. Or, we might be interested in the margin of the classifier with respect to all the data observed (the minimum distance to the decision boundary out of all data seen including the training data and the test point). Our framework will allow for these notions as well, and several of the concrete settings we discuss will use them.

## 1.2 Context and Related Work

**Learning from malicious noise.** The malicious noise model was introduced and analyzed in Valiant [1985], Kearns and Li [1993], Bshouty et al. [2002], Klivans et al. [2009], Awasthi et al. [2017]. See also the book chapter Balcan and Haghtalab [2021]. However, the focus of this work was on the overall error rate of the learned classifier, rather than on instance-wise guarantees that could be provided on individual predictions.

**Instance targeted poisoning attacks.** Instance-targeted poisoning attacks were first introduced by Barreno et al. [2006]. Subsequent work by Suciu et al. [2018] and Shafahi et al. [2018] demonstrated empirically that such attacks can be highly effective, even when the adversary only adds *correctly-labeled data* to the training set (known as "clean-label attacks"). These targeted poisoning attacks have attracted considerable attention in recent years due to their potential to compromise the trustworthiness of learning systems [Geiping et al., 2021, Mozaffari-Kermani et al., 2015, Chen et al., 2017]. Theoretical research on defenses against instance-targeted poisoning attacks has largely focused on developing stability certificates, which indicate when an adversary with a limited budget cannot alter the resulting prediction. For instance, Levine and Feizi [2021] suggest partitioning the training data into $k$ segments, training distinct classifiers on each segment, and using the strength of the majority vote from these classifiers as a stability certificate, as any single poisoned point can affect only one segment. Additionally, Gao et al. [2021] formalize various types of adversarial poisoning attacks and explore the problem of providing stability certificates for them in both distribution-independent and distribution-specific scenarios. Balcan et al. [2022] instead propose correctness certificates: in contrast to the previous results that certify when a budget-limited adversary could not *change* the learner's prediction, their work focuses on certifying the prediction made is *correct*. This model was extended in Balcan et al. [2023] to address test-time attacks as well. The model of Balcan et al. [2022] can be seen as a generalization of the reliable-useful learning framework of Rivest and Sloan [1988] and the perfect selective classification model of El-Yaniv and Wiener [2010], which focus on the simpler scenario of learning from noiseless data, extending it to the more complex context of noisy data and adversarial poisoning attacks.

## 2 Formal Setup

We consider a learner aiming to learn an unknown target function $f^* : \mathcal{X} \to \mathcal{Y}$, where $\mathcal{X}$ denotes the instance space and $\mathcal{Y}$ the label space. The learner is given a training set $S' = \{\{(x_i, y_i)\}_{i=1}^n | x \in \mathcal{X}, y \in \mathcal{Y}\}$, which might have been poisoned by a malicious adversary. Specifically, we assume $S'$ consists of an original dataset $S$ labeled according to $f^*$, with possibly additional examples, whose labels need not match $f^*$, added by an adversary. For original dataset $S$ and non-negative integer $b$, it will be helpful to define $\mathcal{A}_b(S)$ as the possible training sets that could be produced by an attacker with corruption budget $b$. That is, $\mathcal{A}_b(S)$ consists of all $S'$ that could be produced by adding at most $b$ points to $S$. Given the training set $S'$ and test point $x$, the learner's goal will be to output a label $y$ along with a guarantee that $y = f^*(x)$ so long as $f^*$ is sufficiently "simple" and the adversary's corruption budget was sufficiently small. Conceptually, we will imagine that the adversary might have been using its entire corruption budget specifically to cause us to make an error on $x$. Our basic definitions will *not* require that the original set $S$ be drawn iid (or that the test point $x$ be drawn from the same distribution) but our guarantees on the probability mass of points for which a given strength of guarantee can be given will require such assumptions.

**Complexity measures** To establish a framework where certain classifiers or classifications are considered more *natural* than others, we assume access to a *complexity measure* $\mathcal{C}$ that formalizes this degree of unnaturalness. We consider several distinct types of complexity measures.

1. *Data independent*: Each classifier $h$ has a well-defined real-valued complexity $\mathcal{C}(h)$. For example, in $\mathbb{R}^1$, a natural measure of complexity of a Boolean function is the number of alternations between positive and negative regions (See Definition 4.1).

2. *Test data dependent*: Here, complexity is a function of the classifier $h$ and the test point $x_{test}$. For example, suppose $\mathcal{X} = \mathbb{R}^d$ and we allow arbitrary classifiers. If $x_{test}$ is inside a cloud of positive examples, then while there certainly exist classifiers that perform well on the training data and label $x_{test}$ negative, they would necessarily have a small margin with

respect to $x_{test}$. This motivates a complexity measure $\mathcal{C}(h, x_{test}) = \frac{1}{r(x_{test}, h)}$ where $r$ is the distance of $x_{test}$ to $h$'s decision boundary. (See Definition 4.7).

3. *Training data dependent*: This complexity is a function of the classifier $h$ and the training data. An example of this measure is the Interval Probability Mass complexity, detailed in the Appendix (See Definition A.3).

4. *Training and test data dependent*: Here, complexity is a function of the classifier $h$, the training data, and the test point $x_{test}$. For instance, we might be interested in the margin $r$ of a classifier with respect to both the training set and the test point, and define complexity to be $\frac{1}{r}$ (See Definition 4.9).

In section 4, and Appendix A.1, we introduce several complexity measures across all four types, for assessing the structure and behavior of classifiers. We now define the notion of a *regularized-robustly-reliable* learner in the face of instance-targeted attacks. This learner, for any given test example $x_{test}$, outputs both a prediction $y$ and values $c_{low}$ and $c_{high}$, such that $y$ is guaranteed to be correct so long as the target function $f^*$ has complexity less than $c_{high}$ and the adversary has at most corrupted $b$ points. Moreover, there should exist a candidate classifier of complexity at most $c_{low}$.

**Definition 2.1** (Regularized Robustly Reliable Learner). *A learner $\mathcal{L}$ is regularized-robustly-reliable with respect to complexity measure $\mathcal{C}$ if, given training set $S'$, the learner outputs a function $\mathcal{L}_{S'}$ : $\mathcal{X} \times \mathbb{Z}^{\geq 0} \to \mathcal{Y} \times \mathbb{R} \times \mathbb{R}$ with the following properties: Given a test point $x_{test}$, and mistake budget $b$, $\mathcal{L}_{S'}(x_{test}, b)$ outputs a label $y$ along with complexity levels $c_{low}, c_{high}$ such that*

(a) *There exists a classifier $h$ of complexity $c_{low}$ (with respect to $x_{test}$ if test-data-dependent and with respect to some $S$ consistent with $h$ such that $S' \in \mathcal{A}_b(S)$ if training-data-dependent) with at most $b$ mistakes on $S'$ such that $h(x_{test}) = y$, and*

(b) *There is no classifier $h'$ of complexity less than $c_{high}$ (with respect to $x_{test}$ if test-data-dependent and with respect to any $S$ consistent with $h'$ such that $S' \in \mathcal{A}_b(S)$ if training-data-dependent) with at most $b$ mistakes on $S'$ such that $h'(x_{test}) \neq y$.*

*So, if $\mathcal{L}_{S'}(x_{test}, b) = (y, c_{low}, c_{high})$, then we are guaranteed that $y = f^*(x_{test})$ if $S' \in \mathcal{A}_b(S)$ for some true sample set $S \in \mathcal{X} \times \mathcal{Y}$ and $f^*$ has complexity less than $c_{high}$ with respect to $x_{test}$ and $S$.*

**Remark 2.2.** *We define $\mathcal{L}_{S'}$ as taking $b$ as an input, whereas in Balcan et al. [2022], the corruption budget $b$ is an output. We could also define $\mathcal{L}_{S'}$ as taking only $x_{test}$ as input and producing output vectors $\mathbf{y}, \mathbf{c}_{low}, \mathbf{c}_{high}$, where $\mathbf{y}[b]$, $\mathbf{c}_{low}[b]$ and $\mathbf{c}_{high}[b]$ correspond to the outputs of $\mathcal{L}_{S'}(x_{test}, b)$ in Definition 2.1. We define $\mathcal{L}_{S'}$ to take $b$ as an input primarily for clarity of exposition, and all our algorithms indeed can be adapted to output a table of values if desired.*

**Remark 2.3.** *When the learner outputs a value $c_{high} \leq c_{low}$, we interpret it as "abstaining."*

Definition 2.1 motivates the following generic algorithm for implementing a regularized robustly reliable (RRR) learner, for data-independent complexity measures.

---
**Algorithm 1** Generic RRR learner for data-independent complexity measures $\mathcal{C}$
---
1. Given $S'$, find the classifier $h_{S'}$ of minimum complexity that makes at most $b$ mistakes on $S'$.
2. Given test point $x_{test}$, output $(y, c_{low}, c_{high})$ where $y = h_{S'}(x)$, $c_{low} = \mathcal{C}(h_{S'})$, and $c_{high} = \min\{\mathcal{C}(h) : h$ makes at most $b$ mistakes on $S'$ and $h(x) \neq h_{S'}(x)\}$.
---

**Remark 2.4.** *Notice that the generic Algorithm 1 can compute $h_{S'}$ and $c_{low}$ at training time, but requires re-solving an optimization problem on each test example to compute $c_{high}$. (For complexity measures that depend on the test point, even $c_{low}$ may require re-optimizing).*

We now define the notion of a regularized robustly reliable region.

**Definition 2.5** (Empirical Regularized Robustly Reliable Region). *For RRR learner $\mathcal{L}$, dataset $S'$, poisoning budget $b$, and complexity bound $c$, the empirical regularized robustly reliable region $\widehat{\mathrm{R}^4}_{\mathcal{L}}(S', b, c)$ is the set of points $x$ for which $\mathcal{L}_{S'}(x, b)$ outputs $c_{low}, c_{high}$ such that $c_{low} \leq c < c_{high}$.*

Similarly to Balcan et al. [2022], one can characterize the largest possible set $\widehat{\mathrm{R}^4}_{\mathcal{L}}(S', b, c)$ in terms of agreement regions. We describe the characterization below, and prove its optimality in Section 3.

184 **Definition 2.6** (Optimal Empirical Regularized Robustly Reliable Region). *Given dataset $S'$, poi-*
185 *soning budget $b$, and complexity bound $c$, the* optimal empirical regularized robustly reliable region
186 $\widehat{\mathrm{OPTR}}^4(S', b, c)$ *is the agreement region of the set of functions of complexity at most $c$ that make*
187 *at most $b$ mistakes on $S'$. If there are no such functions, then $\widehat{\mathrm{OPTR}}^4(S', b, c)$ is undefined. (For*
188 *data-dependent complexity measures, we define the complexity of a function as its minimum possible*
189 *complexity over possible original training sets $S$, and the point in question if test-data-dependent.)*

Figure 1: The blue regions depict $\widehat{\mathrm{OPTR}}^4(S', 0, 8)$ described in Definition 2.6 for the complexity measure Number of Alternations, mistake budget $b = 0$, and complexity level $c = 8$.

190 In the next section we give a regularized robustly reliable learner $\mathcal{L}$ such that for all $S'$ and $b$, $\mathcal{L}$
191 satisfies $\widehat{\mathrm{R}^4}_{\mathcal{L}}(S', b, c) = \widehat{\mathrm{OPTR}}^4(S', b, c)$ simultaneously for all values of $c$. We then prove that
192 any other regularized robustly reliable learner $\mathcal{L}'$ must have $\widehat{\mathrm{R}^4}_{\mathcal{L}'}(S', b, c) \subseteq \widehat{\mathrm{OPTR}}^4(S', b, c)$. This
193 justifies the use of the term *optimal* in Definition 2.6.

## 3   General Results

195 Recall that a regularized robustly reliable (RRR) learner $\mathcal{L}$ is given a sample $S'$ and outputs a function
196 $\mathcal{L}_{S'}(x, b) = (y, c_{\text{low}}, c_{\text{high}})$ such that if $S' = \mathcal{A}_b(S)$ for some (unknown) uncorrupted sample $S$
197 labeled by some (unknown) target concept $f^*$, and $\mathcal{C}(f^*) \in [c_{\text{low}}, c_{\text{high}})$, then $y = f^*(x)$.

198 **Theorem 3.1.** *For any RRR learner $\mathcal{L}'$ we have $\widehat{\mathrm{R}^4}_{\mathcal{L}'}(S', b, c) \subseteq \widehat{\mathrm{OPTR}}^4(S', b, c)$. Moreover, there*
199 *exists an RRR learner $\mathcal{L}$ such that $\widehat{\mathrm{R}^4}_{\mathcal{L}}(S', b, c) = \widehat{\mathrm{OPTR}}^4(S', b, c)$.*

200 *Proof.* First, consider any $x \notin \widehat{\mathrm{OPTR}}^4(S', b, c)$. This means there exist $h_0$ and $h_1$ of complexity
201 at most $c$, each making at most $b$ mistakes on $S'$, such that $h_0(x) \neq h_1(x)$. In particular, this
202 implies that for any label $y$, there exists a classifier $h'$ of complexity at most $c$ with at most $b$
203 mistakes on $S'$ such that $h'(x) \neq y$. (For data-dependent complexity measures, $h'$ has complexity
204 $c$ with respect to some possible original training set $S$.) So, for any RRR learner $\mathcal{L}'$, by part (b) of
205 Definition 2.1, $\mathcal{L}'$ cannot output $c_{high} > c$, and therefore $x \notin \widehat{\mathrm{R}^4}_{\mathcal{L}'}(S', b, c)$. This establishes that
206 $\widehat{\mathrm{R}^4}_{\mathcal{L}'}(S', b, c) \subseteq \widehat{\mathrm{OPTR}}^4(S', b, c)$.

For the second part of the theorem, let us first consider complexity measures that are not data
dependent. In that case, consider the learner $\mathcal{L}$ given in Algorithm 1 that given $S'$ finds the classifier
$h_{S'}$ of minimum complexity that makes at most $b$ mistakes on $S'$ and then uses it on test point $x$.
Specifically, it outputs $(y, c_{low}, c_{high})$ where $y = h_{S'}(x)$, $c_{low} = \mathcal{C}(h_{S'})$, and

$$c_{high} = \min\{\mathcal{C}(h) : h \text{ makes at most } b \text{ mistakes on } S' \text{ and } h(x) \neq h_{S'}(x)\}.$$

207 By construction, $\mathcal{L}$ is a RRR learner. Now, if $x \in \widehat{\mathrm{OPTR}}^4(S', b, c)$ then this learner $\mathcal{L}$ will output
208 $(y, c_{low}, c_{high})$ such that $c_{low} \leq c$ and $c_{high} > c$. That is because $x$ is in the agreement region of
209 classifiers of complexity at most $c$ that make at most $b$ mistakes on $S'$, which means that any classifier
210 making at most $b$ mistakes on $S'$ that outputs a label different than $y$ on $x$ must have complexity
211 strictly larger than $c$. So, $x \in \widehat{\mathrm{R}^4}_{\mathcal{L}}(S', b, c)$. This establishes that $\widehat{\mathrm{R}^4}_{\mathcal{L}}(S', b, c) \supseteq \widehat{\mathrm{OPTR}}^4(S', b, c)$,
212 which together with the first part implies that $\widehat{\mathrm{R}^4}_{\mathcal{L}}(S', b, c) = \widehat{\mathrm{OPTR}}^4(S', b, c)$.

213 If the complexity measure is data dependent, the learner $\mathcal{L}$ instead works as follows. Given $S'$, $\mathcal{L}$
214 simply stores $S'$ producing $\mathcal{L}_{S'}$. Then, given $x$ and $b$, $\mathcal{L}_{S'}(x, b)$ computes

$$
\begin{aligned}
y &= h_{S'}(x) \text{ where } h_{S'} = \operatorname{argmin}_h\{\mathcal{C}(h, S', b, x) : h \text{ makes at most } b \text{ mistakes on } S'\}, \\
c_{low} &= \mathcal{C}(h_{S'}, S', b, x), \text{ and} \\
c_{high} &= \min\{\mathcal{C}(h, S', b, x) : h \text{ makes at most } b \text{ mistakes on } S' \text{ and } h(x) \neq h_{S'}(x)\},
\end{aligned}
$$

215 where here we define $\mathcal{C}(h, S', b, x)$ as the minimum complexity of $h$ over all possible true training
216 sets $S$, that is, sets $S$ consistent with $h$ such that $S' \in \mathcal{A}_b(S)$. Again, by design, $\mathcal{L}$ is a RRR learner,
217 and if $x \in \widehat{\mathrm{OPTR}}^4(S', b, c)$ then it outputs $(y, c_{low}, c_{high})$ such that $c_{low} \leq c$ and $c_{high} > c$.    $\square$

Definition 2.6 and Theorem 3.1 gave guarantees in terms of the observed sample $S'$. We now consider guarantees in terms of the *original* clean dataset $S$, defining the set of points that the learner will be able to correctly classify and provide meaningful confidence values *no matter how* an adversary corrupts $S$ with up to $b$ poisoned points. For simplicity and to keep the definitions clean, we assume for the remaining portion of this section that $\mathcal{C}$ is *non-data-dependent*.

**Definition 3.2** (Regularized Robustly Reliable Region). *Given a complexity measure $\mathcal{C}$, a sample $S$ labeled by some target function $f^*$ with $\mathcal{C}(f^*) = c$, and a poisoning budget $b$, the regularized robustly reliable region $R_{\mathcal{L}}^4(S, b, c)$ for learner $\mathcal{L}$ is the set of points $x \in \mathcal{X}$ such that for all $S' \in \mathcal{A}_b(S)$ we have $\mathcal{L}_{S'}(x, b) = (y, c_{low}, c_{high})$ with $c_{low} \leq c < c_{high}$.*

**Remark 3.3.** $R_{\mathcal{L}}^4(S, b, c) = \bigcap_{S' \in \mathcal{A}_b(S)} \widehat{\mathrm{R}^4}_{\mathcal{L}}(S', b, c)$.

**Definition 3.4** (Optimal Regularized Robustly Reliable Region). *Given a complexity measure $\mathcal{C}$, a dataset $S$ labeled by some target function $f^*$, with $\mathcal{C}(f^*) = c$, and a poisoning budget $b$, the* optimal regularized robustly reliable region $\mathrm{OPTR}^4(S, b, c)$ *is the agreement region of the set of functions of complexity at most $c$ that make at most $b$ mistakes on $S$. If there are no such functions, then $\mathrm{OPTR}^4(S, b, c)$ is undefined.*

**Theorem 3.5.** *For any RRR learner $\mathcal{L}'$, we have $R_{\mathcal{L}'}^4(S, b, \mathcal{C}(f^*)) \subseteq \mathrm{OPTR}^4(S, b, \mathcal{C}(f^*))$. Moreover, there exists an RRR learner $\mathcal{L}$ such that for any dataset $S$ labeled by (unknown) target function $f^*$, we have $R_{\mathcal{L}}^4(S, b, \mathcal{C}(f^*)) = \mathrm{OPTR}^4(S, b, \mathcal{C}(f^*))$.*

*Proof.* For the first direction, consider $x \notin \mathrm{OPTR}^4(S, b, \mathcal{C}(f^*))$. By definition, there is some $h$ with $\mathcal{C}(h) \leq \mathcal{C}(f^*)$ that makes at most $b$ mistakes on $S$ and has $h(x) \neq f^*(x)$. Now, consider an adversary that adds no poisoned points, so that $S' = S$. In this case, such $h$ makes at most $b$ mistakes on $S'$, as well. Hence, by definition, $c_{\text{high}} \leq \mathcal{C}(f^*)$ and so $x \notin R_{\mathcal{L}}^4(S, b, c)$. Hence, $R_{\mathcal{L}}^4(S, b, c) \subseteq \mathrm{OPTR}^4(S, \mathcal{C}(f^*), b)$. For the second direction, consider a learner $\mathcal{L}$ training set $S'$, finds the classifier $h_{S'}$ of minimum complexity that makes at most $b$ mistakes on $S'$ and then uses it on test point $x$. Specifically, it outputs $(y, c_{\text{low}}, c_{\text{high}})$ where $y = h_{S'}(x)$, $c_{\text{low}} = \mathcal{C}(h_{S'})$, and $c_{\text{high}} = \min\{\mathcal{C}(h) : h \text{ makes at most } b \text{ mistakes on } S' \text{ and } h(x) \neq h_{S'}(x)\}$. By construction, $\mathcal{L}$ satisfies Definition 2.1 and so is a RRR learner. Now, suppose indeed $S' \in \mathcal{A}_b(S)$ for a true set $S$ labeled by target function $f^*$. Then $f^*$ makes at most $b$ mistakes on $S'$, so $\mathcal{L}$ will output $c_{\text{low}} \leq \mathcal{C}(f^*)$. Moreover, if $x \in \mathrm{OPTR}^4(S, f^*, b)$, then any classifier $h$ with $h(x) \neq f^*(x)$ either has complexity strictly greater than $f^*$ or makes more than $b$ mistakes on $S$ (and therefore more than $b$ mistakes on $S'$). Therefore, $\mathcal{L}$ will output $c_{\text{high}} > \mathcal{C}(f^*)$ and have $y = f^*(x)$. So, $x \in R_{\mathcal{L}}^4(S, b, \mathcal{C}(f^*))$. Therefore, $\mathrm{OPTR}^4(S, b, \mathcal{C}(f^*)) \subseteq R_{\mathcal{L}}^4(S, b, \mathcal{C}(f^*))$. $\qquad\square$

**Remark 3.6.** *The adversary's optimal strategy is to add no points, since the learner must consider all classifiers of a given complexity that make at most $b$ mistakes on the training set, and adding new points can only shrink this set.*

# 4 Regularized Robustly Reliable Learners with Efficient Algorithms

In this section, we present efficient algorithms for implementing regularized robustly reliable learners with optimal values of $c_{low}$ and $c_{high}$ for a variety of complexity measures. We present additional examples in the Appendix.

## 4.1 Number of Alternations

We first consider the Number of Alternations complexity measure for data in $\mathbb{R}^1$, and also analyze the sample-complexity for having a large regularized robustly reliable region.

**Definition 4.1** (Number of Alternations). *The number of alterations of a function $f : \mathbb{R} \to \{-1, +1\}$ is the number of times the function's output changes between +1 and -1 as the input variable increases from negative to positive infinity.*

Number of Alternations is a data-independent measure. A higher number of alterations implies a more intricate decision boundary, as the classifier switches between classes more frequently. For instance, if $f$ is the sign of a degree $d$ polynomial, then it can have at most $d$ alterations.

**Example 4.2** (Number of Alterations). *Consider the dataset in Figure 2. Assuming there is no adversary, it is impossible to classify these points with any function that has less than 7 alterations. Suppose we now receive the test point shown in Figure 3. Given a corruption budget b, the learner will output a predicted label and interval $(c_{low}, c_{high})$ as shown in Table 1.*

Table 1: Guarantee for the test point in Figure 3 and the complexity measure Number of Alterations.

| Mistake Budget | Label | $(c_{\text{low}}, c_{\text{high}})$ |
|:---:|:---:|:---:|
| $b = 0$ | $+$ | $[7, 9)$ |
| $b = 1$ | $+$ | $[5, 7)$ |
| $b = 2$ | $+$ | $[3, 5)$ |
| $b = 3$ | $+$ | $[2, 4)$ |
| $b = 4$ | $+$ | $[1, 3)$ |
| $b = 5$ | $+$ | $[1, 2)$ |
| $b = 6$ | Any | $\{1\}$ |
| $b = 7, 8$ | $-$ | $[0, 1)$ |
| $b = 9, 10, 11, 12, 13, 14, 15, 16$ | Any | $\{1\}$ |

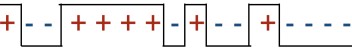

Figure 2: Number of Alterations  Figure 3: Test point arrives

**Definition 4.3** (Optimal Regularized Robustly Reliable Learner). *We say a regularized robustly-reliable learner $\mathcal{L}$ is* optimal *if it outputs values $c_{low}$ and $c_{high}$ that are respectively the lowest and highest possible values satisfying Definition 2.1.*

**Theorem 4.4.** *For binary classification, an optimal regularized-robustly-reliable learner can be implemented efficiently for complexity measure Number of Alterations.*

*Proof sketch.* The high-level idea is to perform bi-directional Dynamic Programming on the training data. A left-to-right DP computes, for each point $i$ and each $j \leq b$, the minimum-complexity solution that makes $j$ mistakes up to that point (that is, on points $0, 1, ..., i$) and labels $i$ as positive, as well as the minimum-complexity solution that makes $j$ mistakes so far and labels $i$ as negative. A right-to-left DP does the same but in right-to-left order. Then, when a test point $x$ arrives, we can use the DP tables to compute the values $y, c_{low}, c_{high}$ in time $O(b)$, without needing to re-train on the training data. In particular, we just need to consider all ways of partitioning the mistake-budget $b$ into $j$ mistakes on the left and $b - j$ mistakes on the right, and then using the DP tables to select the best choice. The full proof is given in Appendix A.2.1. $\qquad\square$

**Remark 4.5.** *If instead of computing $y, c_{low}, c_{high}$ for a single value of $b$ we wish to compute them for all $b \in [0, b_{max}]$, the straightforward approach would take time $O(b_{max}^2)$. However, we can also use an algorithm of Chi et al. [2022] for computing the $(\min, +)$-convolution of monotone sequences to compute the entire set in time $\tilde{O}((b_{max} + c_{max})^{1.5})$, where $c_{max}$ is the largest value in the DP tables (See Theorem A.8 in the Appendix).*

We now analyze the sample complexity for having a large regularlized robustly-reliable region for this complexity measure when data is iid.

**Theorem 4.6.** *Suppose the Number of Alterations of the target function is c. For any $\epsilon, \delta \in (0, 1)$, and any mistake budget $b$, if the size of the (clean) sample $S \sim \mathcal{D}^m$ is at least $\tilde{O}\left(\frac{(b+1)c}{\epsilon}\right)$, and as long as there is at least $\frac{\epsilon}{2c}$ probability mass to the left and right of each alternation of the target function, with probability at least $1 - \delta$, the optimal regularized robustly reliable region, $\mathrm{OPTR}^4(S, c, b)$, contains at least a $1 - \epsilon$ probability mass of the distribution.*

*Proof sketch.* Consider $2c$ intervals $I_1, I_2, ..., I_{2c}$, each of probability mass $\frac{\epsilon}{2c}$ to the left and right of each alternation. Without loss of generality, assume $I_1$ is positive, $I_2$ and $I_3$ are negative, $I_4$ and $I_5$ are positive, etc., according to the target function $f^*$. A sample size of $\tilde{O}(\frac{(b+1)c}{\epsilon})$ is sufficient so that with high probability, $S$ contains at least $b + 1$ points in each of these intervals $I_j$. Assuming $S$

indeed contains such points, then any classifier that does not label at least one point in each interval correctly must have error strictly larger than $b$. This in turn implies that any classifier $h$ with $b$ or fewer mistakes on $S$ must have an alternation from positive to negative within $I_1 \cup I_2$, an alternation from negative to positive within $I_3 \cup I_4$, etc. Therefore, if $h$ has complexity $c$, it *cannot* have any alternations outside of $\bigcup_j I_j$ and indeed must label all of $\mathbb{R} - \bigcup_j I_j$ in the same way as $f^*$. The full proof is given in Appendix A.2.2. $\qquad\square$

## 4.2 Local Margin

We now study a *test-data-dependent* measure.

**Definition 4.7** (Local Margin)**.** *Given a metric space $(\mathcal{M}, d_\mathcal{M})$, for a classifier with a decision function $h : \mathcal{X} \to \mathcal{Y}$, where $\mathcal{X}$ is the input space and $\mathcal{Y}$ is the output space, the local margin of the classifier with respect to a point $x^* \in \mathcal{X}$ is the distance between $x^*$ and the nearest point $x' \in \mathcal{X}$ such that $h(x') \neq h(x^*)$.*

$$r(h, x^*) = \inf_{\{x' \in \mathcal{X} : h(x') \neq h(x^*)\}} d(x^*, x')$$

*We define the local margin complexity measure $\mathcal{C}(h, x^*)$ as $1/r(h, x^*)$.*

A larger local margin implies that the given point is well separated from the decision boundary. For this complexity measure, we have the convenient property that for any training set $S'$, test point $x_{test}$, label $y$, and mistake budget $b$, the minimum complexity $c_{low,y}$ of a classifier $h$ that makes at most $b$ mistakes on $S'$ and gives $x_{test}$ a label of $y$ is given by $1/r$ where $r$ is the distance between $x_{test}$ and the $(b+1)$st closest example in $S'$ of label different from $y$. In particular, $r$ cannot be larger than this value since at least one of these $b + 1$ points must be correctly labeled by $h$ and therefore it is a legitimate choice for $x'$ in Definition 4.7. Moreover, it is realized by the classifier that labels the open ball around $x_{test}$ of radius $r$ as $y$, and then outside of this ball is consistent with the labels of $S'$. This allows us to show:

**Theorem 4.8.** *For any multi-class classification task, an optimal regularized robustly reliable learner can be implemented efficiently for complexity measure Local Margin.*

*Proof sketch.* Given training data $S'$ and test point $x_{test}$, we compute the distance of all training points from $x_{test}$. Then, for each class label $y_i$, we compute the radius $r_i$ of the largest open ball we can draw around the test point that contains at most $b$ training points with label different from $y_i$. The complexity of the least complex classifier that labels the test point as $y_i$ is then $c_{y_i} = \frac{1}{r_i}$. We repeat this for all classes. We then define the predicted label $y = \operatorname{argmin}_{y_i}\{c_{y_i}\}$, $c_{low} = c_y$, and $c_{high} = \min_{y_i \neq y}\{c_{y_i}\}$. An example and the full proof is given in Appendix A.3. $\qquad\square$

## 4.3 Global Margin

Lastly, we study a *test-and-training-data-dependent* measure.

**Definition 4.9** (Global Margin)**.** *Given a metric space $(\mathcal{M}, d_\mathcal{M})$, a set $\tilde{S} = \{(x, y) | x \in \mathcal{X}, y \in \mathcal{Y}\}$, and a classifier $h : \mathcal{X} \to \mathcal{Y}$ that realizes $\tilde{S}$, we define the global margin of $h$ with respect to $\tilde{S}$ as*

$$r(h, \tilde{S}) = \min_{x_i \in \tilde{S}} \inf_{\{x' \in \mathcal{X} : h(x') \neq h(x_i)\}} d(x_i, x').$$

*We define the global margin complexity measure $\mathcal{C}(h, \tilde{S})$ as $1/r(h, \tilde{S})$. Furthermore, given a training set $S'$, test point $x_{test}$ and corruption budget $b$, we define $\mathcal{C}(h, S', b, x_{test})$ as $1/r$ where $r$ is the largest value of $r(h, S \cup \{x_{test}\})$ over all $S$ such that $S' \in \mathcal{A}_b(S)$; that is, it is an "optimistic" value over possible original training sets $S$.*

Intuitively, Global Margin says that the most natural label for a test point $x_{test}$ is the label such that the resulting data is separable by the largest margin. Note that in the presence of an adversary with poisoning budget $b$, the set $\tilde{S}$ in the above definition corresponds to the test point along with the training set $S'$, excluding the $b$ points of $S'$ of smallest margin.

**Theorem 4.10.** *On a binary classification task, an optimal regularized robustly reliable learner can be implemented efficiently for complexity measure Global Margin.*

*Proof sketch.* For simplicity, suppose that instead of being given a mistake-budget $b$ and needing to compute $c_{low}$ and $c_{high}$, we are given a complexity $c$ with associated margin $r = 1/c$ and need to compute the minimum number of mistakes to label the test point as positive or negative subject to this margin. Now, construct a graph on the training data where we connect two examples $x_i, x_j$ if their labels are different and $d(x_i, x_j) < 2r$. Note that the minimum *vertex cover* in this graph gives the smallest number of examples that would need to be removed to make the data consistent with a classifier of complexity $c$. In particular, the nearest-neighbor classifier with respect to the examples remaining (after the vertex cover has been removed) has margin at least $r$, while if a set of examples is removed that is *not* a vertex cover, then the margin of any consistent classifier is strictly less than $r$ by triangle inequality. While Minimum Vertex Cover is NP-hard in general, it is efficiently solvable in *bipartite* graphs via maximum matching, and our graph is bipartite. Now, given our test point $x_{test}$, we can consider the effect of giving it each possible label. If we label $x_{test}$ as positive, then we would want to solve for the minimum vertex-cover *subject to* that cover containing all negative examples within distance $2r$ of $x_{test}$; if we label $x_{test}$ as negative, then we would solve for the minimum vertex cover *subject to* it containing all positive examples within distance $2r$ of $x_{test}$. We can do this by re-solving the maximum matching problem from scratch in the graph in which the associated neighbors of $x_{test}$ have been removed, or we can do this more efficiently (especially when $x_{test}$ does not have many neighbors) by using dynamic algorithms for maximum matching. Such algorithms are able to recompute a maximum matching under small changes to a given graph more quickly than doing so from scratch. Finally, to address the case that we are given the corruption budget $b$ rather than the complexity level $c$, we pre-compute the graphs for all relevant complexity levels and then perform binary search on $c$ at test time. Appendix A.4.1 describes some helpful properties of global margin and A.4.2 contains the proof. □

The above argument is specific to binary classification. We show below that for three or more classes, achieving an optimal regularized robustly reliable learner is NP-hard.

**Theorem 4.11.** *For multi-class classification with $k \geq 3$ classes, achieving an optimal regularized robustly reliable learner for Global Margin complexity is NP-hard.*

*Proof sketch.* We reduce from the problem of Vertex Cover in $k$-regular graphs, which is NP-hard for $k \geq 3$. Given a $k$-regular graph, we first give it a $k$-coloring, which can be done in polynomial time (ignoring the trivial case of the $(k+1)$-clique). We then embed the graph in $\mathbb{R}^m$ such that any two vertices $v_1, v_2$ that were adjacent in the given graph have distance less than $2r$, and any two vertices that were not adjacent have distance greater than $2r$, for some value $r$. The points in this embedding are given labels corresponding to their colors in the $k$-coloring, ensuring that all pairs that were connected in the input graph have different labels. This then gives us that determining the minimum value of $b$ for this radius $r$ is at least as hard as determining the size of the minimum vertex cover in the original graph. The full proof is given in Appendix A.4.3. □

**Other complexity measures** In the appendix, we give regularized robustly reliable learners for other complexity measures including interval probability mass and polynomial degree. We also define the notion of an Empirical Complexity Minimization oracle, analogous to ERM, that computes the general type of optimization needed for achieving an optimal regularized robustly-reliable learner.

# 5 Discussion and Conclusion

In this work, we define and analyze the notion of a *regularized* robustly-reliable learner that can provide meaningful reliability guarantees even for highly-flexible hypothesis classes. We give a generic pointwise-optimal algorithm, proving that it provides the largest possible reliability region simultaneously for all possible target complexity levels. We analyze the probability mass of this region under iid data for the Number of Alternations complexity measure, giving a bound on the number of samples sufficient for it to have large probability mass with high probability. We then give efficient optimal such learners for several natural complexity measures. In the Number of Alternations case, the algorithm uses bidirectional Dynamic Programming to provide its reliability guarantees quickly on new test points without needing to retrain. For Global Margin, we show a reduction to computing maximum matchings in a collection of bipartite graphs and utilize dynamic matching algorithms to produce outputs on test points more quickly than retraining from scratch. A limitation of our work is that in general these guarantees can be very expensive computationally. Nonetheless, we believe our formulation provides an interesting approach to giving meaningful per-instance guarantees for flexible hypothesis families in the face of data-poisoning attacks.

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

# A Empirical Complexity Minimization

**Definition A.1** (Empirical Complexity Minimization). *Given a complexity measure $\mathcal{C}$, a hypothesis class $\mathcal{H}$, a training set $S' = \{(x_1, y_1), (x_2, y_2), ..., (x_n, y_n)\}$, and a mistake budget $b$, let $\mathcal{H}_{b,S'}$ be the set of hypotheses that make at most $b$ mistakes on $S'$:*

$$\mathcal{H}_{b,S'} = \{h \mid \sum_{i=1}^{n} \mathbf{1}[h(x_i) \neq y_i] \leq b\}.$$

*For a data-independent complexity measure, we define the ECM learning rule to choose*

$$h_{ECM} = \arg \min_{h \in \mathcal{H}_{b,S'}} \mathcal{C}(h)$$

*For training-data-dependent complexity measures, we replace $\mathcal{C}(h)$ with the minimum value of $\mathcal{C}(h, \tilde{S})$ over all candidates $\tilde{S}$ for the original training set $S$; that is, $\min\{\mathcal{C}(h, \tilde{S}) : S' \in \mathcal{A}_b(\tilde{S}) \text{ and } h \in \mathcal{H}_{0,\tilde{S}}\}$. When the complexity measure is test-data-dependent (or training-and-test dependent), we define the ECM learning rule to output just the complexity value, rather than a hypothesis.*

$$\min_{h \in \mathcal{H}_{b,S'} : h(x_{test}) = y_{test}} \mathcal{C}(h, x_{test}) \quad or \quad \min_{h \in \mathcal{H}_{b,S'} : h(x_{test}) = y_{test}} \mathcal{C}(h, S', b, x_{test}),$$

*where $\mathcal{C}(h, S', b, x_{test})$ is the minimum value of $\mathcal{C}(h, \tilde{S}, x_{test})$ over all candidates $\tilde{S}$ for the original training set $S$.*

Note that for test-data-dependent complexity measures, an ECM oracle only outputs a complexity value, rather than a classifier, and so would be called for each possible label $y_{test}$, with the algorithm choosing the label of lowest complexity. The reason for this is that typically for such measures, the full classifier itself is quite complicated (e.g., a full Voronoi diagram for nearest-neighbor classification), whereas all we really need is a prediction on $x_{test}$.

## A.1 Other Examples of Complexity Measures

**Definition A.2** (Interval Score). *Let $\{X_1, \ldots, X_n\}$ be a set of $n$ independent and identically distributed real-valued random variables drawn from a distribution $\mathcal{D}$ with cumulative distribution function $F(t)$. The empirical distribution function $\hat{F}_n(t)$ associated with this sample is defined as:*

$$\hat{F}_n(t) = \frac{1}{n} \sum_{i=1}^{n} \mathbf{1}_{\{X_i \leq t\}},$$

*where $\mathbf{1}_{\{X_i \leq t\}}$ denotes the indicator function that is 1 if $X_i \leq t$ and 0 otherwise. Consider $m$ disjoint intervals $I_i = (s_i, e_i]$ on the real line, where $1 \leq i \leq m$. Each interval $I_i$ is associated with a sequence of sample points sharing a common label. The empirical probability mass within an interval $I_i$ is given by:*

$$\hat{F}_n(e_i) - \hat{F}_n(s_i) = \frac{1}{n} \sum_{j=1}^{n} \mathbf{1}_{\{s_i < X_j \leq e_i\}}.$$

*We define the interval score for $I_i$ as:*

$$Score(I_i) = \frac{n}{1 + \sum_{j=1}^{n} \mathbf{1}_{\{s_i < X_j \leq e_i\}}} = \frac{n}{n \cdot \left( \hat{F}_n(e_i) - \hat{F}_n(s_i) + 1 \right)} = \frac{1}{\hat{F}_n(e_i) - \hat{F}_n(s_i) + 1}. \tag{1}$$

In the definition of the score, we add one to the denominator to make sure that every $I_i$ has a non-zero count. This score reflects the inverse of the empirical probability mass contained within the interval $I_i$, and is a *training-data-dependent* measure. A lower mass results in a higher score, indicating that the interval captures a more "complex" region of the sample space. We then define the Interval Probability Mass complexity using Definition A.2 above.

**Definition A.3** (Interval Probability Mass). *The Interval Probability Mass complexity of the set of intervals $\{I_1, \ldots, I_m\}$ is then defined as the aggregate of the interval scores:*

$$Complexity(S) = \sum_{i=1}^{m} Score(I_i) = \sum_{i=1}^{m} \frac{1}{\hat{F}_n(e_i) - \hat{F}_n(s_i) + 1}. \tag{2}$$

Definition A.3 is a training data dependent measure that sums the contributions from all intervals, providing a scalar quantity that quantifies the distribution of the sample points across the intervals. A higher complexity suggests that the sample is dispersed across many low-mass intervals.

**Definition A.4** (Degree of Polynomial). *Let $f(x) = sign[p(x)]$, where $f : \mathbb{R}^n \to \{-1, +1\}$ is defined by a polynomial function $p(x_1, x_2, \ldots, x_n)$ over the input space $\mathcal{X} \subseteq \mathbb{R}^n$, and the function value changes between $+1$ and $-1$ based on the sign of $p(x)$.*

$$p(x) = \sum_{\alpha_1, \alpha_2, \ldots, \alpha_n} c_{\alpha_1, \alpha_2, \ldots, \alpha_n} x_1^{\alpha_1} x_2^{\alpha_2} \ldots x_n^{\alpha_n},$$

*where $\alpha_1, \alpha_2, \ldots, \alpha_n \geq 0$, and $c_{\alpha_1, \alpha_2, \ldots, \alpha_n} \in \mathbb{R}$ are the polynomial coefficients. The degree of the polynomial is defined as the maximum sum of exponents $\alpha_1 + \alpha_2 + \cdots + \alpha_n$ for which the corresponding coefficient is non-zero.*

Degree of Polynomial is a data independent measure. A higher degree indicates more intricate changes in the sign of $f(x)$ across the input space, corresponding to a more complex and flexible boundary. Note that in $\mathbb{R}^1$, the Number of Alternations is a lower bound on the Degree of Polynomial. In Sections A.6 and A.5 we give optimal regularized robustly reliable learners for the Interval Probability Mass and Degree of Polynomial complexity measures, respectively.

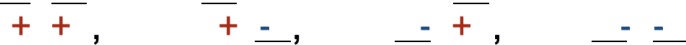

Figure 4: *Illustration of a Function's Behavior on the Left and Right Sides of a Test Point:* **Leftmost:** The function labels both the leftmost and rightmost neighbors of the test point as positive. Labeling the test point as positive does not increase complexity, but labeling it as negative increases the complexity by two. **Middle Figures:** The function labels the left neighbor as positive (or negative) and the right neighbor as negative (or positive). The complexity is the sum of the complexities on each side of the test point plus one, since the function needs to alter in order to connect the left side to the right side, regardless of the test point's label. **Rightmost:** The function labels both neighbors as negative. Labeling the test point as negative does not increase complexity, but labeling it as positive increases the complexity by two.

## A.2 Number of Alterations

### A.2.1 Proof of theorem 4.4

**Theorem 4.4.** *For binary classification, an optimal regularized-robustly-reliable learner (Definition 4.3) can be implemented efficiently for complexity measure Number of Alterations (Definition 4.1).*

*Proof.* Algorithm 2 is the solution. We now prove its correctness. First, we define the DPs that store the scores used, then we use the DP table to compute the complexity level when the test point and mistake budget arrive. We define $DP+, DP-, DP'+, DP'-$ each of which are 2D tables of size $n \times (n+1)$. The rows of the tables denote the position of the current data point, namely for $DP+$ and $DP-$, we denote the rightmost point by index 0, and the leftmost point by index $n - 1$. As for $DP'+$ and $DP'-$, the rows of the tables denote the position of the current data point in the reverse sequence, i.e., we denote the rightmost point by index $n - 1$, and the leftmost point by index 0. The columns of the tables denote the number of mistakes made up to that point which can vary between 0 to the position of the current point+1. We provide the proof of correctness for $DP+$, and it is similar for the other three.

Consider $i = 0$ (the first point in the sequence):

- **If** $a[0] =$ '+':

    - We initialize $DP_+[0][0] = 0$ because the complexity is 0 with no mistakes made, and the rightmost point is positive.
    - We set $DP_+[0][1] = \infty$ since no mistakes can be made yet.

- **If** $a[0] =$ '-':

    - We initialize $DP_+[0][0] = \infty$ because it is impossible to have the rightmost point be positive without making a mistake.
    - We set $DP_+[0][1] = 0$ because removing the negative point gives a valid sequence with complexity 0.

The base case correctly handles both possible labels of the first point, ensuring the initialization aligns with the definition of $DP_+$.

**Induction Hypothesis:** Assume that for all $i' < i$ and all $j$, the table entries $DP_+[i'][j]$ correctly compute the minimum complexity level such that the number of mistakes up to position $i'$ is $j$ and the rightmost existing point in the sequence is positive.

**Inductive Step:** We need to show that $DP_+[i][j]$ is correctly computed for position $i$.

- **Case 1:** $a[i] =$ '+'

    - We have three possible scenarios:
        1. **Keep the point** $a[i]$ **without making a mistake:** This scenario corresponds to $DP_+[i-1][j]$.
        2. **Remove** $a[i]$ **and use** $j - 1$ **mistakes** if the leftmost point is positive: This scenario corresponds to $DP_+[i-1][j-1]$.
        3. **Switch the rightmost point from** $-$ **to** $+$, which adds one to the complexity due to the Alterations: This scenario corresponds to $DP_-[i-1][j] + 1$.
        Thus, the recursive relation is:
        $$DP_+[i][j] = \min(DP_+[i-1][j], DP_+[i-1][j-1], DP_-[i-1][j] + 1)$$
        This relation captures all the valid ways to ensure the rightmost point is positive while maintaining exactly $j$ mistakes.

- **Case 2:** $a[i] =$ '-'

    - To maintain the rightmost point as positive, we must remove $a[i]$, which requires using one of the allowed mistakes:
        $$DP_+[i][j] = DP_+[i-1][j-1]$$
        This equation reflects the necessity to remove a negative point to maintain a valid sequence with a positive rightmost point.

Since the recursive relation properly handles both cases for the current point $i$ based on its label, and the inductive hypothesis ensures correctness for all prior points, the table entry $DP_+[i][j]$ is correctly computed.

**Computing the test label efficiently:** We now use the DP tables to obtain the test label. Note that our approach does not require re-training to compute the test label efficiently.

Once we receive the test point's position along with the adversary's budget, $b$, we compute the *exact* minimum complexity needed to label it point as positive and negative. We denote the test point's position by $test\_pos$, there are four different possibilities for how a function could behave on the left side and the right side of the test point. See figure 4.

Given $b$, we iterate over all possible divisions of mistake budget between the left side and the right side of the test point in each of these four formations. Define the minimum complexity to label the test point as positive, $c_+$, and the minimum complexity to label the test point as negative, $c_-$. Then, $c_{\text{low}} = \min\{c_+, c_-\}$, and $c_{\text{high}} = \max\{c_+, c_-\}$. We output $y_{\text{test}} = \operatorname*{argmin}_{+,-}\{c_+, c_-\}$, along with $c_{\text{low}}, c_{\text{high}}$. $\qquad \square$

**Remark A.5.** *It suffices to run the test prediction with the entire mistake budget, b, since with more deletions the complexity never increases. We use this fact to fill our DP tables as well as do test time computations more efficiently.*

**Remark A.6.** *Theorem 4.4 can be generalized to classification tasks with more than two classes.*

**Definition A.7** ((min, +)-Convolution). *Given two sequences $a = (a[i])_{i=0}^{n-1}$ and $b = (b[i])_{i=0}^{n-1}$, the* (min, +)-*convolution of a and b is a sequence $c = (c[i])_{i=0}^{n-1}$, where*

$$c[k] = \min_{i=0,\dots,k}\{a[i] + b[k-i]\}, \quad for\ k = 0, \dots, n-1.$$

**Theorem A.8.** *Let $a = (a[i])_{i=0}^{n-1}$ and $b = (b[i])_{i=0}^{n-1}$ be two monotonically decreasing sequences of nonnegative integers, where all entries are bounded by $O(n)$. The* (min, +)-*convolution of a and b can be computed in $\tilde{O}(n^{1.5})$ time by reducing the problem to the case of monotonically increasing sequences, which can be solved using the algorithm presented in Theorem 1.2 of Chi et al. [2022].*

*Proof.* The reduction that transforms monotonically decreasing sequences into monotonically increasing sequences is standard; we provide it here for completeness. This reduction allows the application of the efficient algorithm from Chi et al. [2022].

Given the input sequences $a = (a[i])_{i=0}^{n-1}$ and $b = (b[i])_{i=0}^{n-1}$, we first reverse them to obtain:

$$a_{\text{reverse}} = (a[n-1], a[n-2], \dots, a[0]), \quad b_{\text{reverse}} = (b[n-1], b[n-2], \dots, b[0]).$$

The reversed sequences are now monotonically increasing. We then append $n-1$ infinities to both sequences, resulting in:

$$a' = [a_{\text{reverse}}, \infty, \infty, \dots, \infty], \quad b' = [b_{\text{reverse}}, \infty, \infty, \dots, \infty].$$

These transformation steps take $O(n)$ time. Now, we can apply the algorithm from Chi et al. [2022], which computes the (min, +)-convolution of the monotonically increasing sequences in $\tilde{O}(n^{1.5})$ time. Let the result be the sequence $c'$:

$$c'_k = \min_{0 \le i \le k}(a'_i + b'_{k-i}), \quad for\ k = 0, \dots, 2n-2.$$

We claim that removing the first $n$ elements of $c'$ and reversing the remaining sequence yields the desired convolution of the original sequences. Specifically:

- The first $n$ elements of $c'$ represent cases with an excessive mistake budget and should be discarded. For example, $c'[0]$ corresponds to a budget of $2n$, $c'[1]$ to $2n-1$, and so on, down to $c'[n-1]$, which corresponds to $n+1$.

- For indices $k \ge n$, the infinite values in the padded sequences force convolution contributions from lower indices to be ignored, ensuring correctness.

Thus, extracting the last $n$ elements from $c'$ and reversing their order reconstructs the desired convolution of the original decreasing sequences, which completes the proof. $\qquad\square$

### A.2.2 Proof of theorem 4.6

**Theorem 4.6.** *Suppose the Number of Alterations (Definition 4.1) of the target function is c. For any $\epsilon, \delta \in (0, 1)$, and any mistake budget b, if the size of the (clean) sample $S \sim \mathcal{D}^m$ is at least $\tilde{O}\left(\frac{(b+1)c}{\epsilon}\right)$, and as long as there is at least $\frac{\epsilon}{2c}$ probability mass to the left and right of each alternation of the target function, with probability at least $1 - \delta$, the optimal regularized robustly reliable region, $\text{OPTR}^4(S, c, b)$, contains at least a $1 - \epsilon$ probability mass of the distribution.*

*Proof.* We want to make sure with probability at least $1 - \delta$, the optimal regularized robustly reliable region, $\text{OPTR}^4(S, c, b)$, contains at least $1 - \epsilon$ probability mass. Define $2c$ intervals $I_1, I_2, \dots, I_{2c}$, each of probability mass $\frac{\epsilon}{2c}$ to the left and right of each alternation of the target function $f^*$. Without loss of generality, assume $I_1$ is positive, $I_2$ and $I_3$ are negative, $I_4$ and $I_5$ are positive, etc., according

**Algorithm 2** DP Score of Number of Alterations (Definition 4.1)

---
**Input:** $a$: Train set
**Output:** $DP_+, DP_-, DP'_+, DP'_-$
**Function** DpScore($a, b$):

    $n \leftarrow \text{length}(a) \quad a\_reversed \leftarrow \text{reverse}(a)$

    **for** $i \leftarrow 0$ **to** $n$ **do**

        **for** $k \leftarrow 0$ **to** $n-1$ **do**

           $DP_+[i][k], DP_-[i][k], DP'_+[i][k], DP'_-[i][k] \leftarrow \infty$

    **if** $a[0] =\,'+'$ **then**

        $DP_+[0][0] \leftarrow 0$

        $DP_-[0][1] \leftarrow 0$

    **else**

        $DP_+[0][1] \leftarrow 0$

        $DP_-[0][0] \leftarrow 0$

    **if** $a\_reversed[0] =\,'+'$ **then**

        $DP'_+[0][0] \leftarrow 0$

        $DP'_-[0][1] \leftarrow 0$

    **else**

        $DP'_+[0][1] \leftarrow 0$

        $DP'_-[0][0] \leftarrow 0$

    **for** $i \leftarrow 1$ **to** $n-1$ **do**

        **for** $j \leftarrow 0$ **to** $i+1$ **do**

            **if** $a[i] =\,'+'$ **then**

                $DP_+[i][j] \leftarrow \min(DP_+[i-1][j], DP_+[i-1][j-1], DP_-[i-1][j]+1)$

                $DP_-[i][j] \leftarrow DP_-[i-1][j-1]$

            **else if** $a[i] =\,'-'$ **then**

                $DP_-[i][j] \leftarrow \min(DP_+[i-1][j], DP_+[i-1][j-1], DP_+[i-1][j]+1)$

                $DP_+[i][j] \leftarrow DP_+[i-1][j-1]$

            **if** $a'[i] =\,'+'$ **then**

                $DP'_+[i][j] \leftarrow \min(DP'_+[i-1][j], DP'_+[i-1][j-1], DP'_-[i-1][j]+1)$

                $DP'_-[i][j] \leftarrow DP'_-[i-1][j-1]$

            **else if** $a'[i] =\,'-'$ **then**

                $DP'_-[i][j] \leftarrow \min(DP'_+[i-1][j], DP'_+[i-1][j-1], DP'_+[i-1][j]+1)$

                $DP'_+[i][j] \leftarrow DP'_+[i-1][j-1]$

    **return** $DP_+, DP_-, DP'_+, DP'_-$

---

to $f^*$. We will show that a sample size of $\tilde{O}(\frac{(b+1)c}{\epsilon})$ is sufficient so that with high probability, $S$ contains at least $b+1$ points in each of these intervals $I_j$. Assuming $S$ indeed contains such points, then any classifier that does not label at least one point in each interval correctly must have error strictly larger than $b$. This in turn implies that any classifier $h$ with $b$ or fewer mistakes on $S$ must have an alternation from positive to negative within $I_1 \cup I_2$, an alternation from negative to positive within $I_3 \cup I_4$, etc. Therefore, if $h$ has complexity $c$, it *cannot* have any alternations outside of $\bigcup_j I_j$ and indeed must label all of $\mathbb{R} - \bigcup_j I_j$ in the same way as $f^*$. So, all that remains is to argue the sample size bound.

We will use concentration inequalities to derive a bound on the probability that less than $b+1$ points from the sample fall into any of the $2c$ intervals. Let $X_i$ be an indicator random variable such that:

$$X_i = \begin{cases} 1, & \text{if the } i\text{-th sample point falls into interval } I_j, \\ 0, & \text{otherwise.} \end{cases}$$

Thus, the sum $\sum_{i=1}^{m} X_i$ represents the number of sample points in $S$ that fall into interval $I_j$.

The expected number of points in $I_j$, denoted as $\mu$, is given by:

$$\mu = \mathbb{E}\left[\sum_{i=1}^{m} X_i\right] = m \cdot \frac{\epsilon}{2c}.$$

We are interested in the probability that less than or equal to $b+1$ points fall into any of the $2c$ intervals. We use the union bound to ensure that this probability holds across all intervals. That is we will show

$$\mathbb{P}\left(\exists j \text{ such that } \sum_{i=1}^{m} X_i \leq b\right) \leq \delta.$$

To do this, we will prove for a single interval $I_j$:

$$\mathbb{P}\left(\sum_{i=1}^{m} X_i \leq b\right) \leq \frac{\delta}{2c}.$$

Next, we apply Chernoff bounds to control the probability that fewer than $b+1$ points fall into any interval. We are interested in the lower tail of the distribution, and Chernoff's inequality gives us the following bound:

$$\mathbb{P}\left(\sum_{i=1}^{m} X_i \leq \frac{\mu}{2}\right) \leq e^{-\frac{\mu}{8}}.$$

To ensure that this probability is smaller than $\frac{\delta}{2c}$, it suffices to have

$$\mu \geq 8\ln\left(\frac{2c}{\delta}\right).$$

We also need to ensure that the expected number of points in any interval is sufficiently large to account for the threshold $b+1$. Specifically, we need:

$$\mu \geq 2(b+1).$$

Combining both conditions, we require:

$$\mu \geq \max\left\{2(b+1), 8\ln\left(\frac{2c}{\delta}\right)\right\}.$$

$$m \cdot \frac{\epsilon}{2c} \geq 2(b+1) + 8\ln\left(\frac{2c}{\delta}\right).$$

$$m \geq \frac{2c\left(2(b+1) + 8\ln\left(\frac{2c}{\delta}\right)\right)}{\epsilon}.$$

Thus, the sample complexity $m$ is bounded by:

$$m = \tilde{O}\left(\frac{(b+1)c}{\epsilon}\right),$$

Which ensures with high probability $\mathrm{OPTR}^4(S, c, b)$ contains $1-\epsilon$ of the probability mass. Therefore, any test point drawn from the same distribution as $S$, with probability $1 - \epsilon$ belongs to the optimal regularized robustly reliable region. $\qquad\square$

## A.3 Local Margin

**Example A.9** (Local Margin)**.** *Consider the training set $S'$ and test point $x_{test}$ shown in Figure 5. For mistake budget $b = 1$, the local margin of the (dark blue point in the center) test point $(x_{test}, y_{test})$ is 2 if it is labeled as positive, and 1 if it is labeled as negative. Table 6 shows the optimal intervals $(c_{low}, c_{high})$ for all values of $b$.*

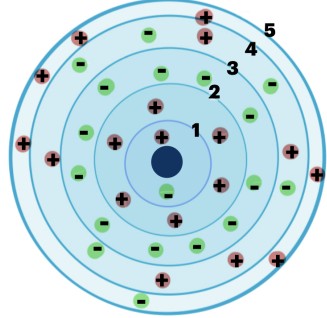

| Mistake Budget | Label | $(c_{\text{low}}, c_{\text{high}})$ |
|---|---|---|
| $b = 0$ | Any | $(3, 3) = \emptyset$ |
| $b = 1, 2, ..., 6$ | $+$ | $[\frac{1}{2}, 1)$ |
| $b = 7, 8, ..., 10, 11$ | $-$ | $[\frac{1}{3}, \frac{1}{2})$ |
| $b = 12, 13, ..., 16$ | $-$ | $[\frac{1}{4}, \frac{1}{3})$ |
| $b = 17$ | Any | $(\frac{1}{4}, \frac{1}{4}) = \emptyset$ |
| $b = 18$ | Any | $(0, 0) = \emptyset$ |

Figure 6: Guarantee for Figure 5.

Figure 5: Local Margin example
($x_{test}$ at center)

As noted in Section 4.2, the lowest-complexity classifier with respect to $(x_{test}, y_{test})$ that makes at most $b$ mistakes on $S'$ has local margin (Definition 4.7) equal to the distance of the test point to the $(b + 1)^{st}$ closest point with a different label. In particular, the margin cannot be larger than this value since at least one of these $b + 1$ points must be correctly labeled by the classifier and therefore it is a legitimate choice for $x'$ in Definition 4.7. Moreover, it is realized by the classifier that labels the open ball around $x_{test}$ of radius this radius as $y_{test}$, and then outside of this ball is consistent with the labels of $S'$.

For example, Table 6 shows the optimal values for the data in Figure 5. So long as the complexity of the target function belongs to the given interval and the adversary has corrupted at most $b$ of the training data points, the given prediction must be correct.

### A.3.1   Proof of Theorem 4.8

**Theorem 4.8.** *For any multi-class classification task, an optimal regularized robustly reliable learner (Definition 4.3) can be implemented efficiently for complexity measure Local Margin (Definition 4.7).*

*Proof.* Given the training data $S'$, the test point $x_{test}$, and the mistake budget $b$, we are interested in the complexity of the classifiers with smallest local margin complexity with respect to the test point and its assigned labels, that make at most $b$ mistakes on $S'$. First, we compute the distance of all training points from the yet unlabeled test point. For each class label, $y_1, y_2, ..., y_m$ create a key in a dictionary and store the distances of all training points (from the test point) with labels opposite to the keys', and sort the values of every key. In a $m$-class classification, there are $m$ keys and each key has at most $n$ entries. The learner starts by labeling the test point as $y_1$, and we check the $y_1$ key in our dictionary. The $b + 1$'th value is the radius of the largest open ball we can draw around the test point labeled as $y_1$ such that it contains at most $b$ points with labels different from $y_1$. We denote this radius by $r_1$. The complexity of the least complex classifier that labels the test point as $y_1$ is $c_{y_1} = \frac{1}{r_1}$. We repeat this for all classes. Without loss of generality, assume $c_{y_1} \leq c_{y_2} \leq \cdots \leq c_{y_k}$. We define:

$$c_{\text{low}} = c_{y_1}, \quad c_{\text{high}} = c_{y_2}$$

where $c_{\text{low}}$ represents the minimum complexity value among the different labelings of $x_{\text{test}}$, and $c_{\text{high}}$ represents the second-lowest complexity value.

Finally, the predicted label for $x_{\text{test}}$ is determined as:

$$y = \underset{y_1, y_2, ..., y_m}{\arg\min} \; \{c_{y_1}, c_{y_2}, \ldots, c_{y_m}\}$$

That is, the label $y$ corresponding to the smallest complexity value is chosen. The learner then outputs the triplet $(y, c_{\text{low}}, c_{\text{high}})$, where $y$ is the predicted label, $c_{\text{low}}$ is the lowest complexity value, and $c_{\text{high}}$ is the second-lowest complexity value, providing a guarantee on the prediction.

$\square$

## A.4 Global Margin

Before proving Theorem 4.10, we first describe some useful properties of the global margin.

### A.4.1 Understanding the Global Margin

Figure 7 shows the margin on one dimensional data. Let $S = \{(x,y)|x \in \mathcal{X}, y \in \mathcal{Y}\}$ denote the set. Given a metric space $(\mathcal{M}, d_{\mathcal{M}})$, draw the largest open ball, $B(x, r_x)$ centered on every $x \in S$, such that for any $(x,y) \in S$, the ball $B(x, r_x)$ does not contain any point $(x', y')$ from the set $S$ with label $y' \neq y$. Each of these balls denotes the (local) margin of their center point. The global margin of the set $S$ is the minimum over radius of such balls.

$$r_S = \min_{x \in S} r_x$$

We now prove the "simplest" classifier, $f^*$, that realizes set $S$ has global margin(Definition 4.9) of $\frac{r_S}{2}$. Moreover, the decision boundary of this classifier must be equidistant between the closest pairs of points with different labels. Hence, the decision boundary is placed midway between the closest points, and the global margin complexity of such function is $\frac{2}{r_S}$.

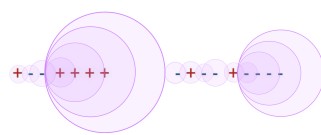

Figure 7: Global Margin on 1-dimensional data. Let $r_S$ be the radius of the smallest ball, and correspond to the distance between the closest pair of points with different labels. Then, the function with minimum global margin complexity with respect to this set is $\frac{2}{r_S}$ complex.

**Theorem A.10.** *Let $(\mathcal{M}, d_{\mathcal{M}})$ be a metric space, and $S = \{(x_i, y_i) \mid x_i \in \mathcal{X}, \ y_i \in \mathcal{Y}\}$ be a finite set of labeled points, where $\mathcal{X}$ is the instance space and $\mathcal{Y}$ is the label space.*

*1. For each $x_i \in \mathcal{X}$, let $r_i$ be the minimum distance from $x_i$ to any point with a different label.*

$$r_i = \inf_{\substack{x_j \in \mathcal{X} \\ y_j \neq y_i}} d_{\mathcal{M}}(x_i, x_j),$$

*2. Let $r_S$ denote the minimum distance between any two differently labeled points in $S$.*

$$r_S = \min_{x_i \in \mathcal{X}} r_i = \min_{\substack{(x_i, y_i), \ (x_j, y_j) \in S \\ y_i \neq y_j}} d_{\mathcal{M}}(x_i, x_j),$$

*Consider a classifier $f^* : \mathcal{X} \to \mathcal{Y}$ that realizes $S$, and obtains minimum global margin complexity (Definition 4.9) with respect to the set $S$. Then the global margin complexity of $f^*$ is $\frac{2}{r_S}$. Moreover, its decision boundary $B_{f^*}$ is placed equidistantly between the closest pairs of points in $S$ with different labels.*

*Proof.* We first show that for any classifier $f^*$ that realizes $S$, the global margin $r$ cannot exceed $\frac{r_S}{2}$. Let $(x_p, y_p), (x_q, y_q) \in S$ be a pair of points such that: $y_p \neq y_q$, and $d_{\mathcal{M}}(x_p, x_q) = r_S$. Since $r_S$ is the minimum distance between any two differently labeled points in $S$, such a pair exists. Consider any classifier $f^*$ that correctly classifies $S$. The minimum distance from $x_p$ (or $x_q$) to the decision boundary cannot exceed $\frac{r_S}{2}$. Formally, since $f^*$ must assign different labels to $x_p$ and $x_q$, there must exist a point $x_b \in B_{f^*}$ such that:

$$d_{\mathcal{M}}(x_p, x_b) + d_{\mathcal{M}}(x_b, x_q) = d_{\mathcal{M}}(x_p, x_q) = r_S.$$

By the triangle inequality, and because $x_b$ lies between $x_p$ and $x_q$, we have:

$$d_{\mathcal{M}}(x_p, x_b) = d_{\mathcal{M}}(x_b, x_q) \geq 0.$$

Since $d_\mathcal{M}(x_p, x_b) + d_\mathcal{M}(x_b, x_q) = r_S$, the maximal possible value for $d_\mathcal{M}(x_p, x_b)$ is $\frac{r_S}{2}$. Therefore, the minimum distance from any point in $S$ to the decision boundary $B_{f^*}$ satisfies:

$$r \leq \frac{r_S}{2}.$$

Now, we construct the classifier $f^*$ (which will just be the nearest-neighbor classifier) that realizes $S$ with a global margin $r = \frac{r_S}{2}$.

Let $f^* : \mathcal{X} \to \mathcal{Y}$ for any $x \in \mathcal{X}$ assign:

$$f^*(x) = \begin{cases} y_i, & \text{if } d_\mathcal{M}(x, x_i) < d_\mathcal{M}(x, x_j) \text{ for all } x_j \in S \text{ with } y_j \neq y_i, \\ y_i \text{ or } y_j, & \text{if } d_\mathcal{M}(x, x_i) = d_\mathcal{M}(x, x_j) \text{ for some } x_j \in S, y_j \neq y_i. \end{cases}$$

This means, place the decision boundary $B_{f^*}$ equidistantly between all pairs $(x_p, y_p), (x_q, y_q) \in S$ with $y_p \neq y_q$ and $d_\mathcal{M}(x_p, x_q) = r_S$. Since $f^*$ assigns to each $x_i \in S$ its correct label $y_i$, it correctly classifies $S$. We will now show that: $r_{f^*} \geq \frac{r_S}{2}$. Assume, for contradiction, that the global margin $r_{f^*} < \frac{r_S}{2}$. Then there exists $x_i \in S$ and $x_b \in B_{f^*}$ such that:

$$d_\mathcal{M}(x_i, x_b) = r - \epsilon < \frac{r_S}{2},$$

for some $\epsilon > 0$. Since $x_b \in B_{f^*}$, there exists $x_j \in S$ with $y_j \neq y_i$ such that:

$$d_\mathcal{M}(x_i, x_b) = d_\mathcal{M}(x_j, x_b).$$

Applying the triangle inequality:

$$d_\mathcal{M}(x_i, x_j) \leq d_\mathcal{M}(x_i, x_b) + d_\mathcal{M}(x_b, x_j) = 2d_\mathcal{M}(x_i, x_b) < r_S.$$

Which contradicts the definition of $r_S$ as the minimum distance between differently labeled points in $S$. Therefore, our assumption is false, and we conclude that:

$$r_{f^*} \geq \frac{r_S}{2}.$$

Combining both directions we get

$$r_{f^*} = \frac{r_S}{2}.$$

$\square$

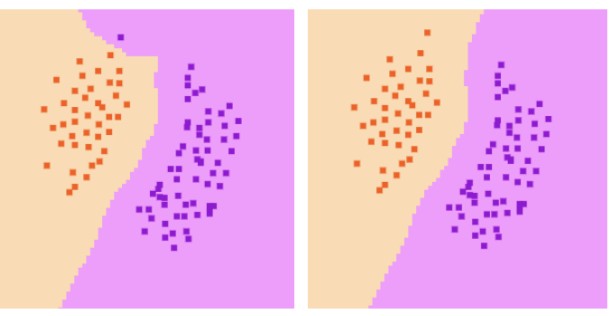

Figure 8: Illustration of Global Margin with different labelings of the test point

### A.4.2 Proof of Theorem 4.10

**Definition A.11** ($(k, r)$-Classification Graph). *Given $S = \{(x, y) | x \in \mathcal{X}, y \in \mathcal{Y}\}$, where $\mathcal{X}$ denotes the instance space and $\mathcal{Y} = \{1, 2, ..., k\}$ the label space, we define the $(k, r)$-**Classification Graph**, $\mathcal{G}_r$, as the graph produced by connecting every two points in $S$ of different labels with distance less than $r$.*

**Remark A.12.** *The Minimum Vertex Cover of $\mathcal{G}_r$ corresponds to the smallest number of points that can be removed from $S$ to make the data consistent with a classifier of global margin complexity $\frac{2}{r}$.*

Using the remark above, we now prove Theorem 4.10.

**Theorem 4.10.** *On a binary classification task, an optimal regularized robustly reliable learner (Definition 4.3) can be implemented efficiently for Global Margin complexity (Definition 4.9).*

*Proof.* Algorithm 4 is the solution. We first compute the distance between every pair of training points, $S'$, with opposite labels. Let $\mathcal{R} = \{0, r_0, r_1, ..., r_p\}$ denote the set of aforementioned distances with an added zero. Without loss of generality, suppose $0 \leq r_0 \leq r_1, ... \leq r_p$. For the case of binary classification, the $(2, r)$-classification graph, $\mathcal{G}_r$, is bipartite. We construct each $(2, r)$-classification graph of the set $\{\mathcal{G}_r(V^+, V^-, E_r)\}_{r \in \mathcal{R}}$ by putting every positive training point in $V^+$, every negative training point in $V^-$, and connecting every two training points of opposite labels with distances less than $r$ by an edge. Since these graphs are bipartite, their Minimum Vertex Cover can be found efficiently by computing a Maximum Matching [Kőnig, 1950]. Notice that by increasing the radius, the Maximum Matching of classification graphs in the set only gets *larger*. Note that there is no edge in $\mathcal{G}_0$; hence the Matching is zero. We continue with computing the Maximum Matching of the classification graph with respect to the smallest radius, $\mathcal{G}_{r_0}$, which corresponds to the largest global margin complexity value. We continue to compute $\{\mathcal{G}_{r_i}\}_{r_i \in \mathcal{R}}$ in ascending order of $i$, and we stop as soon as we reach $p' \in [0, p]$ such that the Maximum Matching of $\mathcal{G}_{r_{p'}}$ is greater than $b$, the mistake budget. Next, when the test point $x_{\text{test}}$ arrives, the learner begins by assigning it a negative label. We compute the distance of the test point, $x_{test}$ from every positive training point. We run a binary search on the possible values of radius, i.e., $[0, p']$. At every level $r_i$, we denote the set of training points labeled as positive with distance less than $r_{i+1}$ from $x_{\text{test}}$ by $\bar{V}_{test}^+$. We denote the cardinality of $\bar{V}_{test}^+$ by $\delta_{test}$, which is indeed the degree of $x_{\text{test}}$ at the current complexity level. If $\delta_{test}$ exceeds our mistake budget, $b$, we break and move to a smaller radius (higher complexity). Otherwise, we add $\delta_{test}$ copies of the test point and connect each of them to a distinct point in $\bar{V}_{test}^+$. We denote the set of $\delta_{test}$ newly added edges by $\bar{E}_{test}$. We have constructed a new graph $\mathcal{G}_{test} = \mathcal{G}_{r_i}(V^+, V^- \cup \{x_{test_i}\}_{i \in [1, \delta_{test}]}, E_{r_i} \cup \bar{E}_{test})$, which ensures all the points adjacent to $x_{test}$ are contained in the Minimum Vertex Cover. We can compute the the Maximum Matching of $\mathcal{G}_{test}$ in time $O(\delta_{test}.(\delta_{test} + |E|))$ by updating the Maximum Matching of $\mathcal{G}_{r_i}$ via computing at most $\delta_{test}$ augmenting paths. Alternatively we can compute the Maximum Matching of $\mathcal{G}_{r_i}$ from scratch in time $O((\delta_{test} + |E|)^{1+o(1)})$ using the fast maximum matching algorithm of Chen et al. [2022]. If the Maximum Matching at the current complexity level exceeds the poisoning budget, $b$, we move to a smaller radius (higher complexity), and if it is less than or equal to our mistake budget, $b$, we search to see if the condition still holds for a larger radius. We accordingly use the corresponding pre-computed representation graphs of the new complexity level. We do the same thing for the test point labeled as positive. Finally, $c_{\text{low}} = \min\{\frac{2}{r_{\max}^+}, \frac{2}{r_{\max}^-}\}$, and $c_{\text{high}} = \max\{\frac{2}{r_{\max}^+}, \frac{2}{r_{\max}^-}\}$. We output $y_{\text{test}} = \underset{+,-}{\mathrm{argmin}}\{\frac{2}{r_{\max}^+}, \frac{2}{r_{\max}^-}\}$, along with $c_{\text{low}}, c_{\text{high}}$. $\square$

**Remark A.13.** *The running time for training-time pre-processing has two main components. The first is construction of the classification graphs. This involves computing all pairwise distances between training points of opposite labels and sorting them; each classification graph $\mathcal{G}_r$ is just a prefix in this list. This portion takes time $O(n^2 \log n)$. The second is computing maximum matchings in each. We can do this from scratch for each graph (Algorithm 3). Alternatively, we can scan the edge list in increasing order, and for each edge insertion just run a single augmenting path (since the maximum matching size can increase by at most 1 per edge insertion). This gives a total cost of at most $O(m^2)$, where $m$ is the number of edges in the graph at the time that the budget $b$ is first exceeded. The running time for test-time prediction is given above, and involves computing at most $\delta_{test}$ augmenting paths per graph in the binary search.*

**Remark A.14.** *The proposed approach is especially fast for small values of $\delta_{test}$, and we can make it faster for large values of $\delta_{test}$, as well. When $\delta_{test}$ is large, one can instead remove $\bar{V}_{test}^+$ vertices from the original graph, $\mathcal{G}_{r_i}$, and re-compute the matching by iteratively finding augmenting paths. We expect the matching of the remaining graph to not exceed $b - \delta_{test}$, and if it does at any step of finding augmenting paths, we can halt. So, the overall time is at most $O((b - \delta_{test}).(\delta_{test} + |E|))$. Alternatively, Bosek et al. [2014] proposed an efficient dynamic algorithm for updating the Maximum Matching of bipartite graphs that can be coupled with our setting and is particularly useful for denser classification graphs, running in time $O((|V^+| + |V^-|)^{3/2})$.*

---

**Algorithm 3** Global Margin (Definition 4.9) Learner Precomputing

---

**Input:** $S$ : Train set, metric $\mathcal{M}$, $b$: Mistake budget
**for** *every* $(x, y), (x', y') \in S'$ *with* $y \neq y'$ **do**
  | Compute $d_{\mathcal{M}}(x, x')$
**end**
Store the sorted distances and zero in $\mathcal{R}_{train} = \{0, r_0, r_1, \ldots, r_{p_{train}}\}$
  Initialize $r \leftarrow 0, p' \leftarrow p_{train}$
  **while** $r \leq p_{train}$ **do**
    **for** *each* $\mathcal{G}_r(V^+, V^-, E_r)$ *where* $r \in \mathcal{R}_{train}$ **do**
      $V^+ \leftarrow \{x \mid (x, y) \in S, \ y = \text{'+'}\}$
      $V^- \leftarrow \{x \mid (x, y) \in S, \ y = \text{'-'}\}$
      $E_r \leftarrow \{e(u, v) \mid u \in V^+, v \in V^-, d_{\mathcal{M}}(u, v) < r\}$
    **end**
    Compute **MaxMatch**$(\mathcal{G}_r)$
    **if** *MaxMatch*$(\mathcal{G}_r) > b$ **then**
      $r_{p'} \leftarrow r - 1$
      **break**
    **end**
    $r \leftarrow r + 1$
**end**
$\mathcal{R}_{train} \leftarrow \{0, r_0, r_1, \ldots, r_{p'}\}$

**return** $\mathcal{R}_{train}, \{\mathcal{G}_r(V^+, V^-, E_r)\}_{r \in \mathcal{R}_{train}}$

---

### A.4.3   Proof of Theorem 4.11

**Definition A.15** (K-Regular Graph). *A graph is said to be $K$-regular if its every vertex has degree $K$.*

**Theorem 4.11.** *For multi-class classification with $k \geq 3$ classes, achieving an optimal regularized robustly reliable learner (Definition 4.3) for Global Margin complexity (Definition 4.9) is NP-hard, and can be done efficiently with access to ECM oracle (Definition A.1).*

*Proof.* We aim to show that finding the minimum VERTEX COVER of a $(k, r)$-representation graph $\mathcal{G}_{(r)}$, for $k \geq 3$ is NP-hard. It is known that finding the VERTEX COVER on cubic graphs is APX-Hard, Alimonti and Kann [2000]. Moreover, by Brooks' theorem, Bona [2016], it is known that a 3-regular graph that is neither complete nor an odd cycle has a chromatic number of 3, and moreover one can find a 3-coloring for such a graph in polynomial time. We now demonstrate that finding the minimum VERTEX COVER for any $k$-colored 3-regular graph, where the graph is neither complete nor an odd cycle, can be reduced in polynomial time to the problem of finding the minimum VERTEX COVER of a $(k, r)$-classification graph. This reduction is accomplished by embedding the vertices of the 3-regular graph into the edge space $\mathbb{R}^m$, where $m = |E|$, the number of edges in the graph. For each vertex $v \in V$, we construct its embedding as follows: if edge $e_i$ is incident to vertex $v$, then the $i$'th dimension of $v$'s embedding is set to 1; otherwise, it is set to 0. Since the graph is 3-regular, each vertex embedding contains exactly three entries of 1, corresponding to the edges incident to that vertex. Finally, each vertex embedding is given a label corresponding to its color in the given $k$-coloring.

The Hamming distance between two vertices in this embedding space encodes adjacency information. Specifically, if two vertices $v_1$ and $v_2$ are adjacent in the graph, their Hamming distance in the embedding space is 4; if they are not adjacent, their distance is 6. This embedding provides a direct correspondence between the adjacency relations in the original graph and the structure of the $(k, r)$-classification graph. Thus, any $k$-colored 3-regular graph can be reduced to a $(k, r)$-classification graph in polynomial time. Given that the VERTEX COVER problem is hard for $k$-regular graphs, it follows that finding the minimum VERTEX COVER in a $(k, r)$-classification graph is also hard. Therefore, implementing the learner $\mathcal{L}$ is NP-hard, completing the proof.

**With ECM Oracle (Definition A.1) Access:** Let $S'$ represent the corrupted training set. To evaluate the test point $x_{\text{test}}$ with label $y_{\text{test}}$, we proceed as follows. First, we augment $S'$ by adding $b + 1$ copies of $x_{\text{test}}$ each labeled as $y_{\text{test}} = y_1$. This ensures that the mistake budget of the ECM algorithm is not

**Algorithm 4** Global Margin (Definition 4.9) Learner

---

**Input:** $x_{\text{test}}$: Test point, $S$: Train set, $b$: Mistake budget, $R_{\text{train}}$: $\{0, r_0, r_1, \ldots, r_{p'}\}$, $\{G_r(V^+, V^-, E_r)\}_{r \in R_{\text{train}}}$

Compute distances from $x_{\text{test}}$ to positive training points.

  Initialize $low \leftarrow 0$, $high \leftarrow |R_{\text{train}}| - 1$, $r_{\max}^+, r_{\max}^- \leftarrow$ None.

  **while** $low < high$ **do**

    Set $mid \leftarrow \lfloor (low + high)/2 \rfloor$

    Set $r_{\text{mid}} \leftarrow R_{\text{train}}[mid]$

    Define $V_{\text{test}}^+ \leftarrow \{p \mid (p, y) \in S, y = \text{`+'}, d_{\mathcal{M}}(p, x_{\text{test}}) < r_{\text{mid}}\}$

    Compute $\delta_{\text{test}} \leftarrow |V_{\text{test}}^+|$

    **if** $\delta_{test} > b$ **then**

      | Set $high \leftarrow mid$ and continue.

    **end**

    Create $\delta_{\text{test}}$ copies of $x_{\text{test}}$, denoted as $\{x_{\text{test},i}\}_{i \in [\delta_{\text{test}}]}$

    **for** $i \in [\delta_{test}]$ **do**

      | Connect $x_{\text{test},i}$ to $V_{\text{test}}^+[i]$ in $G_{r_{\text{mid}}}$

    **end**

    Update Maximum Matching of $G_{r_{\text{mid}}}$

    **if** ***MaxMatch**($G_{r_{mid}}$) $> b$ **then**

      | Set $high \leftarrow mid$.

    **end**

    **else**

      Set $low \leftarrow mid + 1$

      Update $r_{\max}^- \leftarrow R_{\text{train}}[mid - 1]$ if $mid - 1 > 0$, otherwise $r_{\max}^- \leftarrow \min_{p \in V_{\text{test}}^+} d_{\mathcal{M}}(p, x_{\text{test}})$

    **end**

  **end**

**Repeat the above for the negative training points** $(V_{\text{test}}^-, r_{\max}^+)$

**return** $\left( \frac{2}{r_{\max}^+}, \frac{2}{r_{\max}^-} \right)$

---

depleted by the test point $x_{\text{test}}$, as the additional copies force the algorithm to allocate its mistake budget elsewhere.

We then run the ECM algorithm on this modified dataset, and denote the complexity returned by the oracle as $c_{y_1}$. Next, we repeat this procedure for the remaining possible labels $y_2, \ldots, y_m$, each time augmenting the dataset with $b + 1$ copies of $x_{\text{test}}$ labeled according to $y_i$. Let the corresponding complexities returned by the ECM oracle be denoted as $c_{y_2}, \ldots, c_{y_k}$. Without loss of generality, assume $c_{y_1} \leq c_{y_2} \leq \cdots \leq c_{y_k}$ We define:

$$c_{\text{low}} = c_{y_1}, \quad c_{\text{high}} = c_{y_2}$$

where $c_{\text{low}}$ represents the minimum complexity value among the different labelings of $x_{\text{test}}$, and $c_{\text{high}}$ represents the second-lowest complexity value.

Finally, the predicted label for $x_{\text{test}}$ is determined as:

$$y = \operatorname*{argmin}_{y_1, y_2, \ldots, y_k} \{c_{y_1}, c_{y_2}, \ldots, c_{y_k}\}$$

That is, the label $y$ corresponding to the smallest complexity value is chosen. The learner then outputs the triplet $(y, c_{\text{low}}, c_{\text{high}})$, where $y$ is the predicted label, $c_{\text{low}}$ is the lowest complexity value, and $c_{\text{high}}$ is the second-lowest complexity value, providing a guarantee on the prediction.

$\square$

**Example A.16.** *We now aim to demonstrate why such a reduction to the edge space is necessary, and to clarify that not all 3-regular graphs, which are neither complete nor odd cycles, inherently belong to the class of $(k, r)$-Classification Graphs within their original metric space. Consider the well-known Petersen graph, which is a 3-regular and is neither complete nor an odd cycle; hence is 3-colorable. While it satisfies the structural properties for 3-colorability, the graph does not behave as a 3-classification graph when embedded in $\mathbb{R}^2$. Specifically, the metric space properties are not satisfied.*

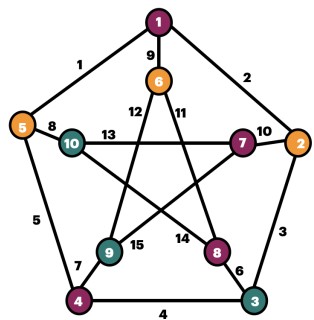

Figure 9: Petersen Graph

*For example, the vertices $v_6$ and $v_{10}$ are closer to each other than the vertices $v_6$ and $v_9$, yet vertices $v_6$ and $v_{10}$ are not connected in the original graph, violating the requirements of a classification graph in its natural embedding. This example highlights that the geometric constraints imposed by the original metric space are too restrictive for certain 3-regular graphs to be used directly as $(k, r)$-classification graphs. To resolve this issue, we embed the vertices of the Petersen graph into the edge space, $\mathbb{R}^m$, where $m = |E|$ is the number of edges in the graph.*

- $v_1 : [1, 1, 0, 0, 0, 0, 0, 0, 1, 0, 0, 0, 0, 0, 0]$,
- $v_2 : [0, 1, 1, 0, 0, 0, 0, 0, 0, 1, 0, 0, 0, 0, 0]$,
- $v_3 : [0, 0, 1, 1, 0, 1, 0, 0, 0, 0, 0, 0, 0, 0, 0]$,
- $v_4 : [0, 0, 0, 1, 1, 0, 1, 0, 0, 0, 0, 0, 0, 0, 0]$,
- $v_5 : [1, 0, 0, 0, 1, 0, 0, 1, 0, 0, 0, 0, 0, 0, 0]$,

- $v_6 : [0, 0, 0, 0, 0, 0, 0, 0, 1, 0, 1, 1, 0, 0, 0]$,
- $v_7 : [0, 0, 0, 0, 0, 0, 0, 0, 0, 1, 1, 0, 0, 0, 1]$,
- $v_8 : [0, 0, 0, 0, 0, 1, 0, 0, 0, 0, 0, 0, 0, 1, 1]$,
- $v_9 : [0, 0, 0, 0, 0, 0, 0, 1, 0, 0, 0, 0, 0, 1, 1, 0]$,
- $v_{10} : [0, 0, 0, 0, 0, 0, 0, 0, 1, 0, 0, 0, 1, 1, 0, 0]$.

*This transformation ensures that the embeddings satisfy the metric space properties required for classification graphs since it preserves the required distance properties for classification: two adjacent vertices in the Petersen graph, such as $v_6$ and $v_9$, have a Hamming distance of 4, while non-adjacent vertices such as $v_6$ and $v_{10}$ have a distance of 6. By embedding the graph into edge space, we transform it into a $(k, r)$-classification graph that respects the desired metric space properties.*

## A.5   Degree of Polynomial

**Theorem A.17.** *On a binary classification task, an optimal regularized robustly reliable learner, $\mathcal{L}$, (Definition 4.3) can be implemented efficiently using ECM oracle (Definition A.1) for complexity measure Degree of Polynomial (Definition A.4).*

*Proof.* Given a corrupted training set $S'$, and a mistake budget $b$, we first run the ECM algorithm on the training set $S'$, which outputs a classifier $h_{S'}$ that minimizes the complexity while making at most $b$ mistakes on $S'$. Let the complexity of $h_{S'}$ be denoted by $c_{\text{low}} = \mathcal{C}(h_{S'})$. The classifier $h_{S'}$ is the minimum complexity classifier among all hypotheses that make no more than $b$ mistakes on $S'$. Using the classifier $h_{S'}$, we label the test point $x_{\text{test}}$, i.e., $y = h_{S'}(x_{\text{test}})$. We modify the training set by adding $b + 1$ copies of the test point $x_{\text{test}}$, but with the label opposite to $y$, i.e., the added points have label $\neg y$. Let this modified set be denoted as $S''$. The addition of $b + 1$ copies of $x_{\text{test}}$ ensures that any classifier produced by ECM will be forced to change the label of $x_{\text{test}}$ if it is to remain within the mistake budget. We now run ECM on the modified training set $S''$, which outputs a new classifier. The complexity of this new classifier is denoted by $c_{\text{high}}$. Since the classifier now labels $x_{\text{test}}$ as $\neg y$, the complexity $c_{\text{high}}$ represents the minimum complexity required to label $x_{\text{test}}$ differently from $h_{S'}(x_{\text{test}})$. By construction, $c_{\text{high}}$ must be greater than or equal to $c_{\text{low}}$ due to the added complexity of labeling the test point differently. Finally, we output the triple $(y, c_{\text{low}}, c_{\text{high}})$ as our guarantee.

$\square$

 **A.6  Interval Probability Mass**

**Definition A.18** (Label Noise Biggio et al. [2011] Adversary). *Label noise was formally introduced in Biggio et al. [2011]. Consider the set of original points $S = \{\{(x_i, y_i)\}_{i=1}^{n} | x \in \mathcal{X}, y \in \mathcal{Y}\}$, where $\mathcal{X}$ denote the instance space and $\mathcal{Y}$ the label space. Concretely, given a mistake budget $b$, the label noise adversary is allowed to alter the labels of at most $b$ points in the dataset $S$. That is, the Hamming distance between the original labels $S$ and the modified labels $S'$, denoted by $d_H(S, S')$, must satisfy the constraint:*

$$d_H(S, S') = \sum_{i=1}^{n} \boldsymbol{I}(y_i \neq y_i' \mid x_i = x_i') \leq b.$$

*Let $\mathcal{A}(S)$ denote the sample corrupted by adversary $\mathcal{A}$. For a mistake budget $b$, let $\mathcal{A}_b$ be the set of adversaries with corruption budget $b$ and $\mathcal{A}_b(S) = \{S' \mid d(S, S') \leq b\}$ denote the possible corrupted training samples under an attack from an adversary in $\mathcal{A}_b$. Intuitively, if the given sample is $S'$, we would like to give guarantees for learning when $S' \in \mathcal{A}_b$ for some (realizable) un-corrupted sample $S$.*

**Theorem A.19.** *For the binary classification task, an optimal regularized robustly reliable learner, $\mathcal{L}$, (Definition 4.3) can be implemented efficiently for complexity measure Interval Probability Mass (Definition A.3) with the label noise adversary (Definition A.18).*

*Proof.* First, we define the DPs that store the scores used, then we use the DP table to compute the complexity level when the test point and mistake budget arrive. We define $DP+, DP-, DP'+, DP'-$ each of which are 3D tables of size $n \times (n+1) \times n$. The first dimension denote the position of the current data point, namely for $DP+$ and $DP-$, we denote the rightmost point by index 0, and the leftmost point by index $n - 1$. As for $DP'+$ and $DP'-$, the first dimension denote the position of the current data point in the reverse sequence, i.e., we denote the rightmost point by index $n - 1$, and the leftmost point by index 0. The second dimension denote the number of mistakes made up to the current point, which can vary between 0 to the number of points so far. Lastly, the third dimension denote the starting point of the interval containing the current point, denoted by the first dimension. We provide the proof of correctness for $DP+$, and it is similar for the other three.

**Base Case** Consider $i = 0$ (the first point in the sequence): Initialize the entire table to infinity.

- **If** $a[0] = $ '+':

  - We initialize $DP_+[0][0][0] = \frac{n}{2}$ because the complexity is $\frac{n}{2}$ with no mistakes made, and the rightmost point is positive.

- **If** $a[0] = $ '-':

  - We set $DP_+[0][1][0] = \frac{n}{2}$, as we can use the mistake budget and flip the negative label to a positive.

**Inductive Hypothesis:** Assume that for all positions up to $i - 1$, the table `DP_+`$[i - 1][j][k]$ correctly stores the minimum complexity score for all possible configurations of mistakes and interval boundaries.

**Inductive Step:** We will show that the table `DP_+`$[i][j][k]$ correctly computes the minimum complexity score at position $i$, based on the following cases:

- **Case 1:** $a[i] = $ '-'

  - **if** $k = i - 1$**:** `DP_+` requires the $i$'th point to be a positive; thus, this point must be removed. We need to decrement the mistake count $j$ of the $i - 1$'th point by one and use it to remove this point. Note that the $i - 1$ must be a negative point in order to have $k = i - 1$.

$$\texttt{DP\_+}[i][j][k] = \min_{k', j' \in [0, j-1]} \left( \texttt{DP\_-}[i-1][j'][k'] \right) + \frac{n}{2}$$

915      **– if** $k < i - 1$**:** Then we flip the label of this point, and update the total score.

$$\texttt{DP\_+[i][j][k]} = \min_{j' \in [0, j-1]} \texttt{DP\_+[i-1][j'][k]} - \frac{n}{i - k + 1} + \frac{n}{i - k + 2}$$

916     • **Case 2:** $a[i] = \text{'+'}$

917      **– if** $k = i - 1$**:** The $i - 1$ must be a negative point in order to have $k = i - 1$.

$$\texttt{DP\_+[i][j][k]} = \min_{k', j' \in [0, j]} (\texttt{DP\_-[i-1][j'][k']}) + \frac{n}{2}$$

918      **– if** $k < i - 1$**:** Then we update the total score.

$$\texttt{DP\_+[i][j][k]} = \min_{j' \in [0, j]} \texttt{DP\_+[i-1][j'][k]} - \frac{n}{i - k + 1} + \frac{n}{i - k + 2}$$

919 Thus, the DP algorithm correctly computes the complexity measure as defined, proving its correctness
920 for `DP_+`.

921 **Computing the test label efficiently:** We now use the DP tables to obtain the test label. Note that our
922 approach does not require re-training to compute the test label efficiently. Once we receive the test
923 point's position along with adversary's budget, $b$, we compute the *exact* minimum complexity needed
924 to label it point as positive and negative. We denote the test point's position by $test\_pos$, there are
925 four different formations for the label of test point's right most and left most neighbor. Given $b$, we
926 iterate over all possible divisions of mistake budget, as well as the position of the starting point of the
927 previous intervals from the left and the right side of the test point in each of these four formations.
928 Define the minimum complexity to label the test point as positive, $c_+$ and the minimum complexity
929 to label the test point as negative, $c_-$. Then, $c_{\text{low}} = \min\{c_+, c_-\}$, and $c_{\text{high}} = \max\{c_+, c_-\}$. We
930 output $y = \underset{+,-}{\text{argmin}}\{c_+, c_-\}$, along with $c_{\text{low}}, c_{\text{high}}$.      □

931 **Remark A.20.** *Theorem A.19 can be generalized to classification tasks with more than two classes.*

**Algorithm 5** DP Score of Interval Probability Mass A.19 with Label Noise A.18

---

**Input:** $a$: Train set
**Output:** $DP_+$, $DP_-$, $DP'_+$, $DP'_-$
**for** $i = 1$ **to** $n$ **do**
    **for** $j = 0$ **to** $i + 2$ **do**
        **for** $k = 0$ **to** $i + 1$ **do**
            **if** $a[i]$ *is* '+' **then**
                **if** $k == i$ **then**
                    $DP_+[i][j][k] \leftarrow \min_{k',j' \in [0,j]}(DP_-[i-1][j'][k']) + \frac{n}{2}$
                    $DP_-[i][j][k] \leftarrow \min_{k',j' \in [0,j]-1}(DP_+[i-1][j'][k']) + \frac{n}{2}$
                **else**
                    $DP_+[i][j][k] \leftarrow \min_{j' \in [0,j]} DP_+[i-1][j'][k] - \frac{n}{i-k+1} + \frac{n}{i-k+2}$
                    $DP_-[i][j][k] \leftarrow \min_{j' \in [0,j-1]} DP_-[i-1][j'][k] - \frac{n}{i-k+1} + \frac{n}{i-k+2}$
                **end**
            **end**
            **if** $a[i]$ *is* '-' **then**
                **if** $k == i$ **then**
                    $DP_+[i][j][k] \leftarrow \min_{k',j' \in [0,j-1]}(DP_-[i-1][j'][k']) + \frac{n}{2}$
                    $DP_-[i][j][k] \leftarrow \min_{k',j' \in [0,j]}(DP_+[i-1][j'][k']) + \frac{n}{2}$
                **else**
                    $DP_+[i][j][k] \leftarrow \min_{j' \in [0,j-1]} DP_+[i-1][j'][k] - \frac{n}{i-k+1} + \frac{n}{i-k+2}$
                    $DP_-[i][j][k] \leftarrow \min_{j' \in [0,j]} DP_-[i-1][j'][k] - \frac{n}{i-k+1} + \frac{n}{i-k+2}$
                **end**
            **end**
            **if** $a\_reversed[i]$ *is* '+' **then**
                **if** $k == i$ **then**
                    $DP'_+[i][j][k] \leftarrow \min_{k',j' \in [0,j]}(DP'_-[i-1][j'][k']) + \frac{n}{2}$
                    $DP'_-[i][j][k] \leftarrow \min_{k',j' \in [0,j-1]}(DP'_+[i-1][j'][k']) + \frac{n}{2}$
                **else**
                    $DP'_+[i][j][k] \leftarrow \min_{j' \in [0,j]} DP'_+[i-1][j'][k] - \frac{n}{i-k+1} + \frac{n}{i-k+2}$
                    $DP'_-[i][j][k] \leftarrow \min_{j' \in [0,j-1]} DP'_-[i-1][j'][k] - \frac{n}{i-k+1} + \frac{n}{i-k+2}$
                **end**
            **end**
            **if** $a\_reversed[i]$ *is* '-' **then**
                **if** $k == i$ **then**
                    $DP'_+[i][j][k] \leftarrow \min_{k',j' \in [0,j-1]}(DP'_-[i-1][j'][k']) + \frac{n}{2}$
                    $DP'_-[i][j][k] \leftarrow \min_{k',j' \in [0,j]}(DP'_+[i-1][j'][k']) + \frac{n}{2}$
                **else**
                    $DP'_+[i][j][k] \leftarrow \min_{j' \in [0,j-1]} DP'_+[i-1][j'][k] - \frac{n}{i-k+1} + \frac{n}{i-k+2}$
                    $DP'_-[i][j][k] \leftarrow \min_{j' \in [0,j]} DP'_-[i-1][j'][k] - \frac{n}{i-k+1} + \frac{n}{i-k+2}$
                **end**
            **else**
            **end**
        **end**
        **end**
    **end**
**end**
**return** $DP_+$, $DP_-$, $DP'_+$, $DP'_-$

---

