# OpenReview forum: "Regularized Robustly Reliable Learners and Instance Targeted Attacks"
_NeurIPS.cc/2025/Workshop/Reliable_ML — NeurIPS 2025 - Reliable ML Workshop_

### Official Review · Reviewer_J94A · 2025-09-12
**Looks like thorough work, not confident enough to give review**

**Rating:** 8
**Confidence:** 1

**Review:**

The subject is not my area of expertise.

---

### Official Review · Reviewer_5giG · 2025-09-13

**Rating:** 7
**Confidence:** 3

**Review:**

**Summary:**

The authors introduce and study the concept of regularized robustly-reliable (RRR) learners. At a high level, given a notion of "classifier complexity", a RRR learner accepts as input a training dataset $S$ containing at most $b$ adversarially corrupted datapoints, and then, given a datapoint $x_{test}$ as input at test time, returns a label $y$ and a complexity level $c$, such that $y$ is guaranteed to be the true label as long as the complexity of the true classifier is at most $c$. The authors provide a generic algorithm for implementing a RRR learner, which they prove is optimal, in the sense that, for any RRR learner, the set of points it labels correctly is a subset of the set of points the optimal RRR learner labels correctly. Then, they provide a series of examples of classifier complexity measures and study whether it is possible to efficiently implement an optimal RRR learner under each one, providing algorithms when possible and a hardness result when not.

**Strengths:**

The paper is well-structured, its main ideas are interesting, and the connections with reliable ML are clear. The fact that the definition of RRR learners can be useful even in cases where the hypothesis class contains classifiers of arbitrary complexity (as seen, for example, when the complexity measure is Number of Alterations and the hypothesis class is all 1-dimensional classifiers) is a solid extension of previous work in robustly-reliable learning. The example complexity measures chosen to illustrate when optimal RRR learners can be efficiently implemented are natural, and demonstrate the usefulness of the RRR framework. All definitions and proofs are adequately explained, the proof sketches are helpful, and the paper is easy to follow.

**Weaknesses/Limitations:**

The paper is somewhat lacking in technical novelty. The results given in section 3 are, for the most part, straightforward implications of the definitions. The example complexity measures, while natural, mostly lend themselves to simple algorithms for efficiently implementing optimal RRR learners; perhaps the only somewhat technically involved results are theorems 4.10 and 4.11. It would have been interesting to see applications of the RRR framework to more complex settings.

Regarding the structure of the paper, there are a few points where it becomes somewhat verbose, for example the generic learner of algorithm 1 is explained again in lines 206-207, 213-216 and 240-244.

**Suggestions for Authors:**

A few observations regarding the structure and formatting of the paper:

1. The fact that the notation for a data-dependent complexity measure is introduced inside a proof in lines 213-217 is somewhat unnatural, maybe the notation for complexity measures could be presented in section 2.

2. While the figures mostly help clarify concepts, there are occasions where they are somewhat unclear and require parsing, particularly in figures 1 and 5.

3. In the proof of theorem 3.5, sometimes $c$ is written instead of $C(f^\star)$, and $OPTR^4(S, C(f^\star), b)$ is written instead of $OPTR^4(S, b, C(f^\star))$.

4. The proof of theorem 4.11 could be cleaned up. It is not explicitly stated what the role of parameter $r$ is in the reduction, and also, the construction of the embedding seems redundant; given a 3-regular instance of Vertex Cover, it seems to me that the corresponding $(3, r)$-classification graph is constructed just by 3-coloring the instance.

---

### Official Review · Reviewer_W2sr · 2025-09-14
**The paper has a strong theoretical contribution with limited empirical validation, but well-suited for the workshop**

**Rating:** 7
**Confidence:** 3

**Review:**

Summary:
This paper addresses the problem of instance-targeted data poisoning attacks by extending the formula proposed by Balcan et al(2022). They address two limitations of the original algorithm: (i) the vacuity of the definition for flexible hypothesis classes, and (ii) the inefficiency of retraining per test point. The authors introduce regularized robustly reliable learners (RRR), parameterized by complexity measures. They prove optimality results for the RRR region,  and propose efficient algorithms for specific measures like number of alternations, local margin and global margin.

Strengths:
- The introduction of regularization via complexity measures is a meaningful generalization that avoids vacuity and provides richer per-instance guarantees.
- The theory is solid and strong. Formal definitions, optimality theorems and sample complexity bounds are well-developed.
- Efficient implementations for several complexity measures are developed.

Weakness:
- The work is entirely theoretical. Probably several experiments or figures, worked examples would help understanding.
- "Number of Alternations" and other measures are mathematically neat but detached from modern ML practice. More discussion on extension to linear models, kernels or deep nets would improve impact.

Suggestions
- Add some examples or figures showing how RRR regions look would be helpful understanding.
- If possible, add some experience demonstrating that RRR learners provide meaningful guarantees.

---

### Official Review · Reviewer_9x7C · 2025-09-20

**Rating:** 7
**Confidence:** 2

**Review:**

This paper extends robustly-reliable learning by introducing regularized robustly-reliable learners, which resolve vacuity issues for expressive hypothesis classes and provide more efficient test-time guarantees. The theoretical contributions—including optimality, sample complexity, and efficient algorithms—are elegant and well-structured. However, the lack of empirical validation and reliance on abstract complexity measures limit immediate practical relevance. Still, the ideas are strong and aligned with the workshop theme.